# Homeostatic control of deep sleep and molecular correlates of sleep pressure in *Drosophila*

**Budhaditya Chowdhury[1†], Lakshman Abhilash[1†], Antonio Ortega[2], Sha Liu[2], Orie Shafer[1]\***

[1]The Advanced Science Research Center, The City University of New York; The Graduate Center at the City University of New York, New York, United States; [2]VIB-KU Leuven Center for Brain & Disease Research, Leuven, Belgium

**Abstract** Homeostatic control of sleep is typically addressed through mechanical stimulation-induced forced wakefulness and the measurement of subsequent increases in sleep. A major confound attends this approach: biological responses to deprivation may reflect a direct response to the mechanical insult rather than to the loss of sleep. Similar confounds accompany all forms of sleep deprivation and represent a major challenge to the field. Here, we describe a new paradigm for sleep deprivation in *Drosophila* that fully accounts for sleep-independent effects. Our results reveal that deep sleep states are the primary target of homeostatic control and establish the presence of multi-cycle sleep rebound following deprivation. Furthermore, we establish that specific deprivation of deep sleep states results in state-specific homeostatic rebound. Finally, by accounting for the molecular effects of mechanical stimulation during deprivation experiments, we show that serotonin levels track sleep pressure in the fly's central brain. Our results illustrate the critical need to control for sleep-independent effects of deprivation when examining the molecular correlates of sleep pressure and call for a critical reassessment of work that has not accounted for such non-specific effects.

## Editor's evaluation

This important study provides a significant methodological advance for the study of *Drosophila* sleep, especially with regard to the study of its homeostatic features, as well as in its reevaluation of the 5-min rest period that is currently used to define the sleep state in *Drosophila*. Although time will tell whether the findings reported survive the test of time, this creative and imaginative piece of work provides solid strengths of evidence and food for thought, as well as important technical developments for the field.

## Introduction

Sleep is a fundamental biological need and is homeostatically controlled: when sleep is disrupted an elevated drive to sleep (i.e., sleepiness) is produced that is associated with decreased performance, well-being, and safety (*Dijk and von Schantz, 2005*; *Holding et al., 2020*; *Krause et al., 2017*; *Lane et al., 2023*; *Meyer et al., 2022*). Sleep disruption is widespread in the modern age and contributes significantly to a wide array of negative health consequences in humans (*Colten and Altevogt, 2006*; *Medic et al., 2017*). It is therefore critical to understand the homeostatic regulatory mechanisms of sleep. Sleep is ubiquitous in the animal kingdom and sleep-like states likely evolved at least 740 million years ago (*Campbell and Tobler, 1984*; *Freiberg, 2020*; *Kanaya et al., 2020*; *Keene and*

**\*For correspondence:**
oshafer@gc.cuny.edu

[†]These authors contributed equally to this work

**Competing interest:** The authors declare that no competing interests exist.

*Duboue, 2018*; *Nath et al., 2017*; *Webb, 1974*). Being so deeply conserved, mechanisms of sleep regulation uncovered in animals with relatively simple nervous systems can be highly relevant to sleep in mammalian species, including humans. The relative simplicity of invertebrate nervous systems and the existence of invertebrate models for which there are well-established methods of genetic and neuronal manipulations make them amenable to experimental approaches that would be technically and logistically difficult or prohibitively expensive in mammalian model systems. Work in such model systems is therefore an important complement to studies in mammals and other vertebrate models (*Davis and Raizen, 2017*).

The vinegar fly *Drosophila melanogaster* has proved to be a useful model organism for understanding the molecular and cellular mechanisms regulating sleep (*Hendricks et al., 2000*; *Shaw et al., 2000*). The powerful genetic tools available in flies, along with the ability to manipulate and measure neural signaling within small groups of defined cell types, have revealed genes and brain networks that mediate the homeostatic regulation of sleep (*DeJesúsOlmo et al., 2020*; *Donlea et al., 2014*; *Ni et al., 2019*; *Seidner et al., 2015*). Furthermore, such work in the fly has revealed mechanisms that also regulate sleep in mammals (*Liu et al., 2016*; *Shafer and Keene, 2021*). Just as in humans and other mammalian species, fly sleep is controlled by both a homeostat and a circadian clock, which, during normal sleep cycles, largely account for the timing and amount of daily sleep (*Abhilash and Shafer, 2023*; *Borbély, 1982*; *Borbély et al., 2016*; *Deboer, 2018*). Future work in the fly will undoubtedly continue to enrich our understanding of sleep regulation in all animals.

Nevertheless, there are significant challenges to the investigation of sleep regulation in *Drosophila*. For example, the magnitude of sleep rebound in flies is quite modest (*Huber et al., 2004*; *Klose and Shaw, 2021*; *Oh et al., 2014*; *Vaccaro et al., 2020*). Indeed, some forms of optogenetic or thermogenetic sleep deprivation produce no apparent homeostatic sleep rebound at all (*Seidner et al., 2015*; *Vaccaro et al., 2020*). As for other animal models of sleep, the action of the fly's sleep homeostat is assayed by depriving flies of sleep and measuring subsequent increases of sleep (compared to baseline), which reflect a homeostatic discharge of the sleep debt accrued during deprivation (*Hendricks et al., 2000*; *Huber et al., 2004*; *Melnattur et al., 2020*; *Shaw et al., 2000*). In contrast to mammalian species, however, sleep rebound following deprivation is often quite modest and brief. For example, male flies display variable levels of increased sleep following nighttime sleep deprivation of around 0–25 min, compared to 200–500 min of sleep lost (in 6 and 12 hr deprivation windows) and this is usually detectable only during the first few hours of the day after deprivation has ended (*Geissmann et al., 2019*; *Geissmann et al., 2017*; *Hendricks et al., 2000*; *Huber et al., 2004*; *Klose and Shaw, 2021*). Furthermore, homeostatic sleep rebound in *Drosophila* appears to repay a relatively small proportion of the sleep lost during deprivation; 800 min of sleep loss in a full diurnal cycle was followed by only 100 min of recovery sleep (~12.5%) (*Huber et al., 2004*). In contrast, mammalian species tend to display higher amounts of increased sleep following deprivation that is observable over more than one circadian cycle (*Berger and Oswald, 1962*; *Mistlberger et al., 1983*; *Nakazawa et al., 1978*; *Tobler et al., 1983*; *Williams et al., 1964*), and such rebound usually accounts for a much larger proportion of lost sleep than the rebound detectable in flies. For example, humans can reclaim around 40% of the sleep lost to deprivation (*Stroemel-Scheder and Lautenbacher, 2021*). The relatively small rebound measured in the fly is a major challenge to the study of sleep regulation in this species, as it provides a small dynamic range over which to detect changes in homeostatic responses in experimental subjects. These features of fly sleep are somewhat surprising, given the highly conserved nature of sleep in the animal kingdom.

One possible cause of the relatively small homeostatic responses to sleep deprivation in flies are the sleep-independent effects of the deprivation techniques used by the field. The most commonly used method of fly sleep deprivation is frequent mechanical perturbation, typically shaking or slamming (*Hendricks et al., 2000*; *Melnattur et al., 2020*; *Shaw et al., 2000*), to force movement and prevent the attainment of sleep, which in *Drosophila* is defined as any bout of inactivity that is 5 min or longer (*Hendricks et al., 2000*; *Shaw et al., 2000*). Though effective for the prevention of sleep, these methods are likely to produce many physiological and behavioral effects that are independent of sleep loss. Indeed, mechanical shaking is known to produce the biochemical hallmarks of stress, even when delivered during times of wakefulness (*Harbison and Sehgal, 2009*).

Recently, neurogenetic methods have been employed to deprive flies of sleep (*Dubowy et al., 2016*; *Vaccaro et al., 2020*). This method involves the strong and chronic excitation of wake-promoting

neurons in the brain. Though this method avoids the sleep-independent effects of physical perturbation, the strong non-physiological excitation of central brain neurons is likely to be attended by its own sleep-independent effects. A fundamental challenge, therefore, is to differentiate sleep-pressure-driven changes in behavior and physiology from sleep-independent changes driven by the manipulation used to prevent sleep. We predict that accounting for such sleep-independent effects would improve the sensitivity with which homeostatic sleep responses can be detected in the fly.

In mammals, sleep consists of multiple, physiologically discrete stages, which differentially respond to sleep deprivation (*Achermann et al., 1993*; *Berger and Oswald, 1962*; *Endo et al., 1997*; *Everson et al., 1989*; *Loomis et al., 1937*; *Rechtschaffen et al., 1999*; *Takahashi et al., 1978*). Following long bouts of wakefulness during normal sleep cycles, slow-wave non-rapid-eye-movement (NREM) sleep stages dominate the initial bouts of sleep, suggesting that the daily rise of sleep pressure is discharged first by slow-wave (i.e., deep) sleep (*Borbély and Achermann, 1999*; *Webb, 1974*). Slow-wave sleep is also preferentially increased following sleep deprivation when compared to rapid eye movement (REM) sleep (*Berger and Oswald, 1962*; *Borbly and Neuhaus, 1979*; *Dijk et al., 1990*; *Williams et al., 1964*). Thus, differentiating between distinct sleep stages is likely important for assessing the homeostatic responses to sleep deprivation.

A growing body of evidence indicates the existence of a deep sleep state in flies, which is characterized by distinct patterns of brain signaling (*Nitz et al., 2002*; *van Alphen et al., 2013*), reduced metabolic rate (*Stahl et al., 2017*), and waste clearance from the brain (*van Alphen et al., 2021*). The physiological and metabolic correlates of this deep sleep state are associated with bouts of inactivity that are significantly longer than the 5 min inactivity criterion used to define fly sleep (*Nitz et al., 2002*; *Stahl et al., 2017*; *van Alphen et al., 2021*). Furthermore, recent studies have suggested that periods of inactivity briefer than the 5 min criterion may represent an 'active sleep' state in the fly (*Anthoney et al., 2023*; *Tainton-Heap et al., 2021*). The existence of multiple sleep states in the fly suggests that treating sleep as a unitary state in this species might obscure homeostatic sleep responses, particularly if, as in mammals, specific states more strongly linked to homeostatic sleep pressure than shallower stages of sleep. Furthermore, our recent analyses suggest that longer bouts of sleep are a better reflection of sleep homeostasis than traditional definitions of sleep in the fly (*Abhilash and Shafer, 2023*).

In this study we address the sleep-independent behavioral effects of the most commonly used form of sleep deprivation in the fly and develop a method of accounting for them. We also attempt to address the extent to which the treatment of sleep as a single unitary state in the fly might explain the apparently modest and brief nature of sleep rebound in the fly compared to such rebound in mammalian species. We present evidence that the extent of mechanical perturbation employed during sleep deprivation has significant effects on the amount of subsequent sleep displayed by deprived flies. We describe the development of a yoked control paradigm for flies, based on previous work in rats (*Rechtschaffen et al., 1983*), that allows us to produce two sets of flies that have experienced identical levels of mechanical perturbation while suffering significantly different amounts of sleep restriction. Furthermore, by differentiating between long bouts of sleep from the traditional unitary definition of sleep, we show that flies display significant and lasting homeostatic sleep increases following sleep deprivation that are only detectable when controlling for the sleep-independent effects of mechanical deprivation. Finally, we illustrate the importance of yoked controls for examining the molecular correlates of sleep pressure and identify serotonin as a molecule that is increased within the brain in proportion to sleep loss but not mechanical perturbation. Our work introduces methodological approaches that are likely to support the discovery of new mechanisms of sleep regulation in the fly, calls for the reevaluation of previous work identifying the molecular, physiological, and cellular correlates of sleep pressure, and suggests that serotonin may act as a sleep substance within the central brain.

## Results
### Sleep rebound is significantly shaped by the frequency of the mechanical stimulus used to prevent sleep
The drive to sleep in the face of prolonged sleep deprivation is so strong that significant experimental intervention is necessary to keep animals awake. Such intervention is known to produce

sleep-independent changes in behavior and physiology in addition to changes caused by increased sleep pressure (*Nollet et al., 2020*). Traditionally, sleep-deprived flies, which have undergone forced wakefulness by means of mechanical or neurogenetic perturbation, are compared to unperturbed flies (e.g., *Hendricks et al., 2000*). Thus, it is not possible to differentiate between sleep-dependent and sleep-independent effects of deprivation, as the experimental flies have experienced both sleep deprivation and experimental insults (typically, physical agitation), whereas control flies have experienced neither. A further complication in assessing sleep rebound is the duration and time of day during which forced wakefulness stimuli are applied (*Huber et al., 2004*) as both levels of sleep pressure and diurnal/circadian changes in physiology can have complex interactions with the perturbing stimuli.

Though the minimum criterion for a sleep-like state in *Drosophila* is 5 continuous minutes of inactivity, the most common means of mechanical sleep deprivation in the field consists of the delivery of mechanical stimulation (shaking or slamming) for 2 s randomly within every 20 s interval (*Geissmann et al., 2019*; *Hendricks et al., 2000*; *Shaw et al., 2000*). Once the permissive window of sleep opportunity is open after the period of sleep deprivation, usually timed to the beginning of the day, sleep is measured across next few hours and the amount of sleep displayed by sleep-deprived flies is compared to unperturbed controls (*Beckwith et al., 2017*; *Geissmann et al., 2019*; *Geissmann et al., 2017*). The amount of baseline sleep displayed by sleep-deprived and control flies before deprivation is also often accounted for in the assessment of rebound to control for small but significant differences in baseline sleep that are often observed between experimental and control flies, even when the two groups are genetically identical (*Geissmann et al., 2019*; *Figure 1A*). Given the substantial discrepancy between the inactivity duration criterion for sleep (5 min or more) and the frequency of mechanical stimulation used to deprive flies of sleep (multiple times per minute) we first asked how rebound sleep might differ between flies that were sleep deprived with different frequencies of mechanical stimulation. Therefore, we varied the duration of mechanical deprivation to compare our results with previous deprivation studies, which have employed various durations of sleep deprivation (*Huber et al., 2004*; *Nall and Sehgal, 2013*; *Seidner et al., 2015*).

As previously described in multiple studies by others using the *Drosophila* Activity Monitor (DAM) system, flies deprived of sleep for 6, 12, and 24 hr using 2 s of mechanical stimulation delivered at random times within each 20 s interval displayed immediate increases in total sleep compared to undisturbed flies (*Figure 1A, D and G*; *Cirelli et al., 2005*; *Hendricks et al., 2000*; *Huber et al., 2004*). In each of these instances a significant 24 hr rebound was observed when compared to undisturbed flies (*Figure 1C, F, I*). This reproducibility with previous studies confirmed that our lab population of *Canton-S* flies display homeostatic sleep increases like those described in previous studies (*Hendricks et al., 2000*; *Huber et al., 2004*; *Melnattur et al., 2020*; *Nall and Sehgal, 2013*; *Shafer and Keene, 2021*; *Shaw et al., 2000*). To ask how the extent of mechanical stimulation shapes homeostatic sleep rebound, we decreased the frequency of such stimulation while maintaining the three durations (6 hr, 12 hr, and 24 hr) of sleep deprivation, and ensuring effective prevention of sleep. In order to examine the effects of stimulation frequency on sleep rebound, we decreased the mechanical stimulation frequency to once every 220 s. This low-frequency trigger produced sleep deprivation for all the deprivation durations tested (*Figure 1B, E, and H*). Surprisingly, no sleep rebound was observed for 6 hr, 12 hr, or 24 hr deprivation experiments, despite the fact that the lower frequency stimulations effectively deprived sleep (*Figure 1C, F, I*). Lowering the intensity of mechanical stimulation therefore appeared to eliminate sleep rebound, despite having prevented sleep during deprivation.

Examination of the magnitudes of sleep rebound following high-frequency mechanical deprivation revealed that the 6 hr deprivation protocol produced higher sleep rebound than did 12 hr and 24 hr deprivation, despite the fact that this duration produced the smallest loss of sleep (*Figure 1J*). To further confirm the role of mechanical disturbance in recovery sleep assessment, we added an additional trigger frequency of random shaking for 2 s within every 120 s interval for 24 hr of sleep deprivation. We compared the sleep rebound produced by 24 hr of sleep deprivation with stimulation frequencies of 20 s, 120 s, and 220 s using a two-way ANOVA and found that there was a significant interaction effect of treatment × trigger frequency on the amount of sleep rebound (total sleep post-deprivation minus total sleep pre-deprivation). Remarkably, we found significant rebound only for the flies deprived with 20 s trigger frequencies (*Figure 1K*). It is also important to note that although the

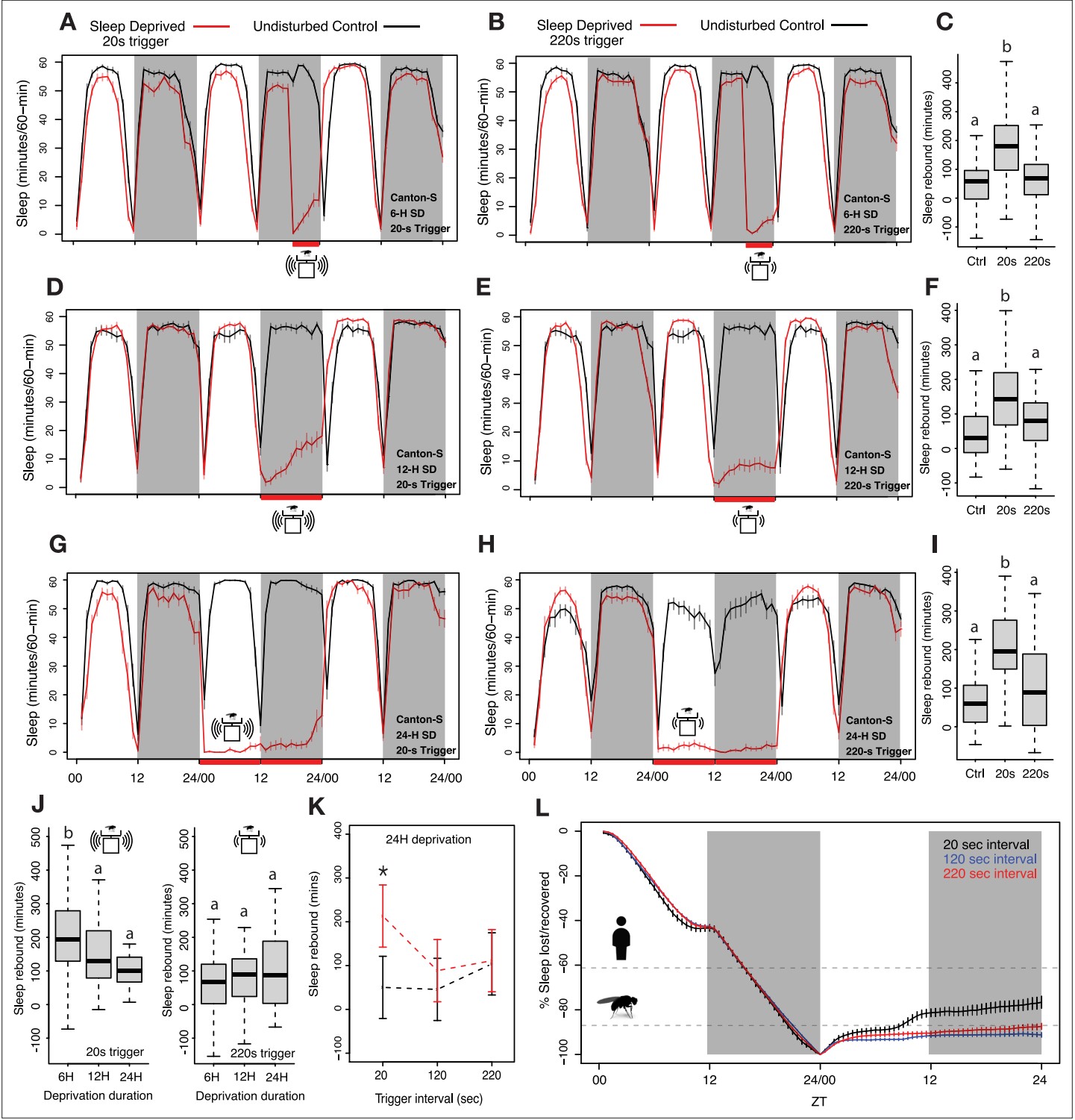

**Figure 1.** Homeostatic sleep rebound is driven by the frequency of mechanical agitation when the amount of sleep deprivation is held constant. (**A–B**) Six hours of sleep deprivation was induced with two different mechanical trigger frequencies using vortexers: 2 s randomized shaking every 20 s or 220 s (each producing similar levels of sleep deprivation). (**C**) A significant increase in sleep was observed for the 20 s trigger, but the 220 s trigger failed to show such increases when compared with undisturbed controls (*Kruskal-Wallis critical value* = 40.43, df = 2, p=1.66 × 10⁻⁹, followed by Bonferroni corrections for pairwise comparisons). (**D–E**) Time course of 12 hours of nighttime sleep deprivation using two trigger frequencies (20 s and 220 s) results in nighttime sleep loss. (**F**) Sleep rebound using 20 s trigger produced significant increases compared to controls, but we did not observe any significant changes when the 220 s trigger was used (*Kruskal-Wallis critical value* = 24.59, df = 2, p=4.5 × 10⁻⁶, followed by Bonferroni corrections for pairwise comparisons). (**G–H**) We increased the vortexing-induced sleep deprivation to 24 hours using 20 s and 220 s trigger frequencies and calculated sleep

*Figure 1 continued on next page*

*Figure 1 continued*

rebound in the 24 hours post-deprivation window represented in the time course. (**I**) Once again, the 20 s trigger produced significant sleep rebound compared to control flies whereas the 220 s trigger-induced deprivation did not, even though both trigger frequencies effectively deprived sleep (*Kruskal-Wallis critical value* = 23.63, df = 2, p=7.41 × 10$^{-6}$, followed by Bonferroni corrections for pairwise comparisons). (**J**) We compared the amount of sleep rebound during the first recovery day caused by different durations of deprivation using 20 s (left) and 220 s (right) inactivity triggers and found significant rebound only for the 20 s trigger. However, despite producing significantly different amounts of sleep loss, we observed no increases in rebound with the 220 s trigger. Surprisingly, the 6 hr deprivation condition, which produced the least amount of sleep loss displayed significantly higher rebound sleep compared to the 12 hr and 24 hr deprivation conditions (20 s trigger: *Kruskal-Wallis critical value* = 13.71, df = 2, p=0.0011, followed by Bonferroni corrections for pairwise comparisons; 220 s trigger: *Kruskal-Wallis critical value* = 1.96, df = 2, p=0.37, followed by Bonferroni corrections for pairwise comparisons). (**K**) We included an intermediate trigger frequency of 120 s for sleep deprivation and found that like the 220 s trigger, it failed to produce statistically significant increases in sleep for the 24 hr post-deprivation analysis (ANOVA $F_{2,250}$=3.93, p=0.02). (**L**) Sleep levels observed during the 24 hr following deprivation reveal that flies recover very little of total sleep lost. Previously measured *Drosophila* (*Huber et al., 2004*) and human sleep (*Stroemel-Scheder and Lautenbacher, 2021*) recovery amounts are indicated by dashed lines. 6 hr SD control n=44, sleep-deprived$_{20 s}$ n=62, sleep-deprived$_{220 s}$ n=57; 12 hr SD control n=32, sleep-deprived$_{20 s}$ n=64, sleep-deprived$_{220 s}$ n=64; 24 hr SD control n=64, sleep-deprived$_{20 s}$ n=32, sleep-deprived$_{220 s}$ n=64, sleep-deprived$_{120 s}$ n=64; *<0.05 and NS (not significant).

The online version of this article includes the following figure supplement(s) for figure 1:

**Figure supplement 1.** Time courses of 'active sleep' – bouts of inactivity that are between 1 and 5 min long – under conditions of 6 (**A**), 12 (**B**), and 24 hr (**C**) sleep deprivation using vortexers at a trigger frequency of 220 s for *CS* flies.

higher trigger frequency produced a significant rebound, the amount of recovered sleep was very small (*Figure 1L*).

Thus, lower frequencies of mechanical stimulation failed to produce significant sleep rebounds, despite having produced similar extents of sleep deprivation. To address the extent to which our sleep-deprived flies recovered lost sleep, we estimated how much lost sleep was recovered across the 24 hr following the end of the deprivation window (*Donlea et al., 2014*; *Shaw et al., 2002*). We found that, as expected from previous work in many labs, the flies deprived of sleep using the most frequent mechanical stimulation (every 20 s) displayed apparent homeostatic sleep increases in the hours immediately following deprivation (*Figure 1A, D, and G*). In contrast, flies that experienced less frequent stimulation displayed little if any such apparent rebound (*Figure 1B, E, and H*). The fact that lower levels of mechanical stimulation failed to produce rebound over one full cycle, despite having produced similar amounts of sleep deprivation, suggests that high-frequency physical perturbation produces sleep-independent behavioral effects that might mask bona fide homeostatic sleep increases. Alternatively, it is also possible that during lower stimulation frequencies, flies may increase their levels of the recently hypothesized 'active sleep' state (inactivity of between 1 and 5 min in duration associated with physiological changes) (*Anthoney et al., 2023*; *Tainton-Heap et al., 2021*) and that this may prevent the build-up of sleep pressure and rebound. We examined the levels of such sleep during and after deprivation using 220 s stimulation frequencies and found that the flies do display substantial increases in this short sleep state throughout deprivation using 220 s stimulation frequencies (*Figure 1—figure supplement 1*). However, given the strong convergent evidence that periods of inactivity of 5 min or more represent a physiologically distinct sleep-like state, it is surprising to see the absence of rebound following deprivation of this sleep state in our experiments using 220 s stimulation frequencies and standard sleep definitions. The absence of rebound in these experiments supports the hypothesis that the apparent homeostatic rebound of fly sleep is strongly shaped by the frequency of mechanical stimulation. Accounting for the sleep-independent effects of mechanical deprivation on sleep and examining the homeostatic control of distinct sleep states are likely to reveal previously hidden characteristics of sleep homeostasis.

## Accounting for mechanical stimulus using a yoked-controlled design uncovers significant sleep rebound in sleep-deprived flies when low-frequency triggers are used

The mechanical shaking employed to sleep-deprived flies loaded into the DAM system uses near-constant agitation without regard to the flies' sleep state, delivering mechanical stimulation to both active and inactive flies. To further examine the potential sleep-independent effects of mechanical perturbation during sleep deprivation experiments, we adopted the Ethoscope, a video-based system for recording fly behavior (*Geissmann et al., 2017*). In contrast to the DAM system, which requires

the simultaneous mechanical perturbation of all flies within the same DAM, the Ethoscope allows the user to deliver mechanical stimuli to individual flies only when a fly is inactive for a specified amount of time (*Geissmann et al., 2017*). The ability to track single flies and selectively stimulate individual flies to prevent sleep means that flies can be sleep deprived with significantly fewer mechanical perturbations, and only when flies have been inactive for defined durations.

We programmed Ethoscopes to track single flies and rotate their tubes at ~420 rpm for 1 s each time a fly had been inactive for 220 s, which resulted in 24 hr of sleep deprivation when using the standard inactivity criterion for fly sleep (5 min or more) (*Figure 2A and C*). For both *Canton-S* and *w1118*, when compared to undisturbed flies, the sleep-deprived flies failed to display significant increases in sleep during the 24 hr following deprivation (*Figure 2B and D*); a surprising result given the clear effectiveness of sleep deprivation (*Figure 2A and C*), but consistent with the lack of rebound in similar situations using the vortexer (see *Figure 1*). This result is also surprising on another account – the use of Ethoscopes to measure sleep does not overestimate and saturate sleep as much as the IR-beam crossing-based method in the DAMs (compare *Figures 1 and 2*), thereby providing a larger dynamic range wherein rebound could have been observed.

The failure to produce clear homeostatic sleep increases following deprivation using a 220 s inactivity criterion could be the product of two causes. As suggested by our observations above, the mechanical insult experienced by flies during sleep deprivation are likely to exert sleep-independent effects on the behavior observed after deprivation. That is, the mechanical disturbance experienced by the sleep-deprived flies may produce effects on the fly that could obscure sleep-pressure-driven behavioral changes. Second, the 220 s inactivity triggers used to deprive flies of sleep may have allowed brief inactivity bouts recently hypothesized to represent an 'active sleep' state (*Anthoney et al., 2023*; *Tainton-Heap et al., 2021*) and such a state prevented the build-up of sufficient sleep pressure to produce a homeostatic response. This would suggest that the 5 min inactivity criterion for sleep may be too long to capture all forms of homeostatically controlled fly sleep. To begin to address these two possibilities, we first considered the effects of mechanical perturbation and therefore sought to produce flies that differed in the extent of their sleep deprivation, despite experiencing identical mechanical stimulation. Comparing sleep in such flies would allow us to examine the specific contribution of sleep pressure to sleep behavior following deprivation.

To accomplish this, we adopted a paradigm of 'yoked' sleep deprivation that was first introduced in the rat model, which produces pairs of rats that have experienced perfectly matched mechanical stimulation, in the form of a rotating platform over water, that nevertheless differed significantly in the amount of sleep deprivation experienced (*Rechtschaffen et al., 1983*). This approach involved a 'focal' animal which was placed on one side of the platform and monitored via electroencephalogram recording. Whenever the focal animal displayed the physiological hallmarks of sleep, the platform would begin to rotate, thereby keeping the focal animal awake. A paired rat, the 'yoked' control, was placed on the opposite side of the platform and could sleep when the focal rat was awake. This arrangement produced a pair of rats that experienced identical, time-matched mechanical stimulation while experiencing significantly different amounts of sleep loss (*Rechtschaffen et al., 1983*).

We programmed the Ethoscope platform to produce a paired yoked control fly for every sleep-deprived focal fly. We conducted these experiments for a 24 hr period commencing at lights-on to avoid the circadian confounds of mechanical stimulations given at specific times of the day or night. For each fly that was tracked and stimulated upon being inactive for 220 s, a second fly received matched, time-locked tube rotations, which were delivered independently of its sleep/wake state. This ensured that each focal and yoked pair received identical mechanical stimuli (*Figure 2E*). 220 s inactivity criteria were necessary to provide sufficient time for our yoked controls to attain substantial amounts of sleep, as defined by the standard criterion of 5 min or more of inactivity. Indeed, using the 220 s inactivity trigger allowed yoked flies to attain sleep while the focal fly was active (*Figure 2F and G*). Yoked controls, therefore, suffered only partial sleep loss compared to focal flies (*Figure 2H and J*). This approach, therefore, succeeded in producing two sets of flies that had experienced the same level of physical perturbation while suffering significantly different levels of sleep loss.

A comparison of total sleep across the entire 24 hr of the post-deprivation day revealed significantly higher levels of sleep in the focal flies compared to yoked controls for both *Canton-S* and *w1118* (*Figure 2I and K*). Thus, the inclusion of yoked controls revealed a homeostatic sleep response that was not apparent when comparing sleep-deprived flies to unperturbed controls, supporting

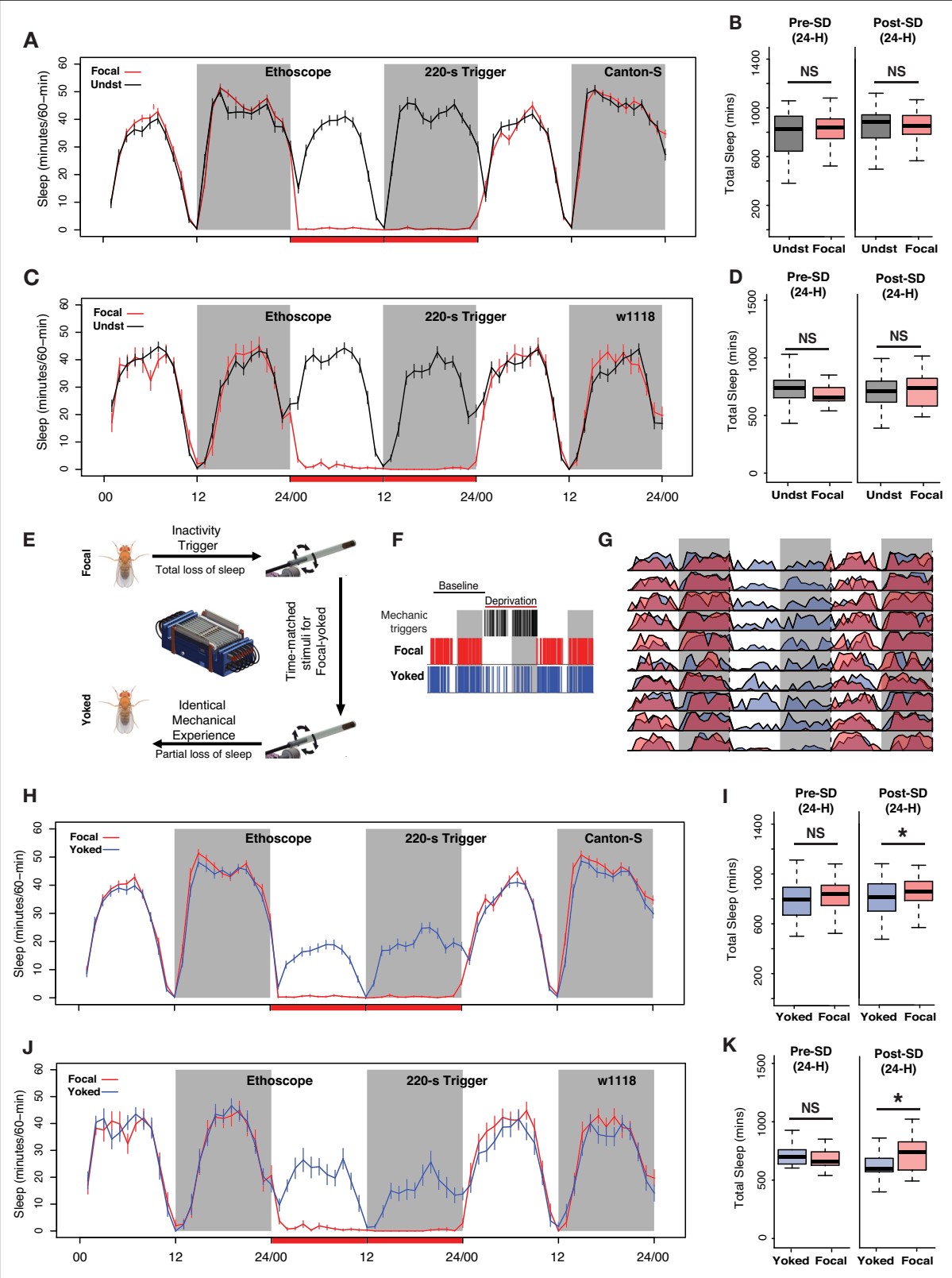

**Figure 2.** Controlling for mechanical stimulation is necessary to observe 24 hr sleep rebound when low-frequency mechanical triggers are used for deprivation. (**A**) Inactivity-dependent sleep deprivation in Ethoscopes with 220 s inactivity triggers fails to generate post-deprivation sleep rebound in wild-type (*Canton-S*) flies when compared with undisturbed controls. (**B**) Focal flies do not show differences in total sleep compared to undisturbed flies across the baseline day (Wilcoxon's W=2523, p=0.54) or across the first post-deprivation day (Wilcoxon's W=2340, p=0.87). (**C**) *w1118* flies also

*Figure 2 continued on next page*

*Figure 2 continued*

failed to display post-deprivation sleep rebound compared to undisturbed controls. (**D**) Comparison of total sleep on the day before and after sleep deprivation for $w^{1118}$ flies. There were no significant differences in sleep amount on the baseline day (baseline sleep: Wilcoxon's $W=351$, $p=0.07$) or on the post-deprivation day (post-deprivation sleep: Wilcoxon's $W=519$, $p=0.71$). (**E**) The Ethoscope platform was modified to include 'yoked controls' where inactivity-dependent sleep depriving triggers for experimental (focal) flies are matched to their yoked controls, thereby generating paired flies with identical mechanical experience but significantly different amounts of sleep loss. (**F**) Sleep and mechanical perturbation in an example of a focal/yoked pair of flies: Sleep episodes for one baseline day and the day of deprivation are shown. Sleep episodes are indicated by red (focal) and blue (yoked) hash marks. Black hashmarks represent tube rotations triggered by the focal fly (left panel). (**G**) Averaged sleep profiles for 10 focal-yoked paired flies on baseline, deprivation, and post-deprivation days show how yoked flies (blue) were able to sleep during the deprivation cycle, whereas focal (red) flies lost all sleep. (**H**) Average traces of focal and yoked flies 24 hr before, during, and after sleep deprivation for wild-type *Canton-S* flies. (**I**) When compared to yoked flies that have undergone same levels of mechanical perturbation, focal flies increased sleep after 24 hr deprivation (Wilcoxon's $W=4074.5$, $p=0.01$) while not showing any baseline differences on the day before deprivation (Wilcoxon's $W=3829$, $p=0.12$). (**J**) To confirm reproducibility of yoking we used the $w^{1118}$ line, a common genetic background strain, to replicate the sleep rebound finding observed for *Canton-S*. (**K**) Compared to yoked flies that have undergone same levels of mechanical perturbation, focal flies show increased sleep following 24 hr deprivation (post-deprivation sleep: Wilcoxon's $W=277$, $p=0.037$) while not showing any baseline differences on the day before deprivation (baseline sleep: Wilcoxon's $W=176$, $p=0.53$). For *Canton-S*, n=82 each for focal and yoked categories, and n=58 for unperturbed controls. For $w^{1118}$, n=20 for focal and yoked categories, and n=49 for unperturbed controls *<0.05 and NS (not significant).

The online version of this article includes the following figure supplement(s) for figure 2:

**Figure supplement 1.** Mechanical perturbation causes long term locomotor effects.

---

the notion that sleep-independent effects of mechanical perturbation can mask homeostatic sleep responses. Both *Canton-S* (*Figure 2H and I*) and $w^{1118}$ (*Figure 2J and K*) strains displayed such rebound, exhibiting increases in both daytime and nighttime sleep during the 24 hr following deprivation. This observation, however, did not answer why sleep-deprived flies fail to show rebound sleep when compared to undisturbed flies. After all, sleep was restricted successfully in both cases of experimental flies. We hypothesized that the mechanical stimulus itself must be inducing unknown, non-specific effects that directly affects long-term sleep or activity. Indeed, we found that mechanical stimulation during deprivation significantly increased the activity levels of sleep-deprived flies a day after mechanical perturbation, when compared to undisturbed controls. This increase was seen immediately after the light-to-dark transition, that is, at the time associated with the transition to nighttime sleep (*Figure 2—figure supplement 1A and B*), and most likely masked the 24 hr sleep rebound when compared to undisturbed flies. This difference was not observed when comparing focal flies with yoked controls (*Figure 2—figure supplement 1C and D*), because both of these paired flies received identical mechanical stimuli. Though yoked controls revealed significant sleep-pressure-specific increases in sleep, this homeostatic increase was quite modest, resulting in very little discharge of the sleep pressure built during the 24 hr of sleep deprivation (*Figure 2I and K*). Given that most animals display homeostatic sleep increases over multiple sleep/wake cycles (*Mistlberger et al., 1983*; *Nakazawa et al., 1978*; *Rechtschaffen et al., 1999*), we wondered if the rebound we detected using yoked controls might persist over multiple diurnal cycles.

## Sleep rebound accumulates over multiple cycles in response to infrequent mechanical deprivation, but little sleep debt is paid off

The circadian control of sleep thresholds ensures that sleep is gated to occur at the appropriate/adaptive time of the day, even in the presence of increased sleep drive. The action of the circadian system therefore prevents sleep debt from being discharged completely upon the first opportunity to sleep, ensuring that wakefulness will occur during the next diurnal cycle despite the presence of homeostatic sleep pressure. For this reason, sleep debt is typically discharged over several sleep/wake cycles (*Mistlberger et al., 1983*; *Nakazawa et al., 1978*; *Rechtschaffen et al., 1999*). Current methods in the *Drosophila* sleep field do not typically produce multi-cycle rebound, but instead, reveal fairly modest sleep rebound during the first hours of the daytime following deprivation (*Beckwith et al., 2017*; *Geissmann et al., 2019*; *Geissmann et al., 2017*). In our yoked-controlled experiment, we observed only modest sleep rebound that discharged very little of the sleep lost during the previous day's deprivation, which may have been a product of the smaller but significant deprivation experienced by yoked controls. However, in this case, sleep rebound was apparent across the diurnal cycle, rather than being limited to the first few hours of the recovery day (*Figure 2H and J*). We wondered

if the comparison of deprived flies with yoked controls might reveal persistent increases in sleep for deprived flies across subsequent sleep/wake cycles. We therefore assessed the behavior of flies deprived of sleep using the Ethoscope for three full diurnal cycles following 24 hr of sleep deprivation (*Figure 3A and B*) and compared the sleep of deprived flies with sleep observed in both undisturbed and paired yoked controls.

When compared to yoked controls, focal sleep-deprived flies displayed modest but significant increases in cumulative sleep over two successive cycles (*Figure 3D*). Once again, these increases were not detectable when deprived flies were compared to undisturbed controls (*Figure 3C*). In this case, baseline sleep on the days before deprivation was not different between the three groups of flies (see *Figure 2B and G*). We examined how much lost sleep was recovered across the three post-deprivation cycles by quantifying sleep gain in focal flies, which we calculated as the increase in total post-deprivation sleep compared to baseline sleep and normalized to sleep in undisturbed controls to account for normal changes in sleep during the duration of the experiment (see Methods). Despite this apparent two-cycle rebound, focal flies failed to discharge a substantial portion of the sleep debt accrued during deprivation (*Figure 3E*). This result is surprising because when we estimated p(Doze), a recently introduced conditional probability metric for sleep pressure (*Wiggin et al., 2020*), we found that the focal flies show sustained increases in sleep pressure for at least 3 days post sleep deprivation (*Figure 3—figure supplement 1*).

We wondered if the apparent absence of substantial homeostatic payback of sleep lost by focal flies during deprivation might be explained by the fact that current sleep analysis methods do not recognize distinct sleep states, treating sleep as a unitary state in which all bouts of inactivity of 5 min or more are considered to be the same sleep state. Given the growing evidence of physiologically and metabolically distinct sleep states in the fly (*Stahl et al., 2017*; *van Alphen et al., 2021*) and the fact that deep sleep stages are most strongly and immediately affected by sleep deprivation in mammals, we turned our attention to how the architecture of sleep might differ between focally deprived flies and their yoked controls across the 3 days following deprivation. Though the number of sleep bouts was not significantly different between focal and yoked flies across any of the three post-deprivation cycles (*Figure 3—figure supplement 2A*), there was a significant increase in bout duration in the focal flies compared to yoked controls across the first 2 days following deprivation (*Figure 3—figure supplement 2B*). These two groups of flies displayed no differences in these metrics during baseline sleep before deprivation (*Figure 3—figure supplement 2C*). This result suggested that focal flies, which had higher sleep pressures than yoked controls, displayed increased sleep bout durations.

## The magnitude of homeostatic sleep rebound in the fly is masked by short-bout sleep

Recent work in our lab revealed that long-bout sleep (~30 min or more of inactivity) is a better reflection of sleep homeostat action than shorter bouts of inactivity (*Abhilash and Shafer, 2023*). Furthermore, such long-bout sleep has been shown by others to reflect a deep sleep stage that is physiologically and metabolically distinct from shorter bouts of sleep (*Stahl et al., 2017*; *van Alphen et al., 2021*). We therefore hypothesized that the extent of sleep rebound and payback of lost sleep might be more readily apparent if we focused our analysis on long-bout sleep alone, eliminating the contribution of shorter, presumably shallower, bouts of sleep.

We reanalyzed the 24 hr sleep deprivation experiment reported in *Figure 3*, iteratively varying the inactivity duration criterion for sleep from 1 to 30 min, a range, which includes short sleep epochs (defined as any bout of inactivity lasting 1–5 min), to ask how varying the sleep definition might shape the apparent homeostatic recovery of lost sleep. We found that inactivity criteria of 25 min or more resulted in the largest payback of lost sleep when examined on the third day of recovery from deprivation (*Figure 4B*). This suggested that longer bouts of sleep are most strongly influenced by homeostatic sleep regulation and that sleep bouts lasting 1–24 min are characterized by substantially lower rebound following deprivation. We found no evidence of post-deprivation increases in brief, short sleep epochs (periods of inactivity lasting between 1 and 5 min). This was not surprising given our finding that 220 s inactivity triggers produce increases in such short sleep bouts during deprivation (*Figure 1—figure supplement 1*). When we employed an inactivity criterion of 25 min, a duration that likely reflects a relatively deep sleep state (*Stahl et al., 2017*) sleep recovery approached ~50% of lost

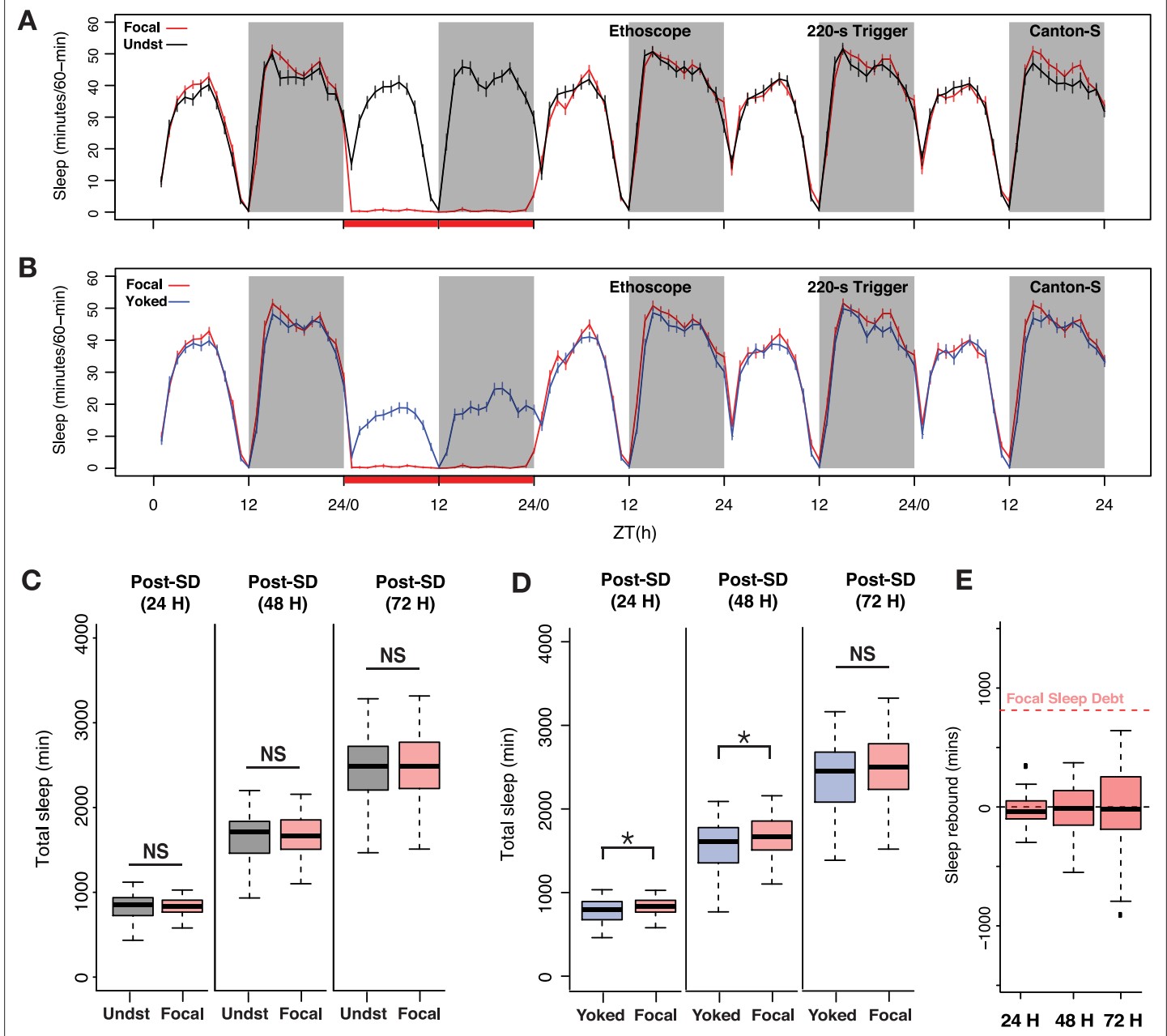

**Figure 3.** Sleep pressure is discharged as cumulative sleep recovery across multiple cycles. (**A**) Average sleep time courses of baseline, deprivation (220 s inactivity-dependent trigger), and 3 post-deprivation days in focal and undisturbed flies, and (**B**) focal-yoked pairs for the same days shown in A (the time courses visualized here are extended, multi-day analysis of the focal-yoked experiment reported for *Canton-S* in the previous figure). (**C**) Focal flies do not show significant differences in cumulative sleep gain across 3 post-deprivation days when compared to undisturbed controls (post 1: Wilcoxon's $W=2323.5$, $p=0.81$; post 2: Wilcoxon's $W=2351$, $p=0.91$; post 3: Wilcoxon's $W=2460.5$, $p=0.72$). (**D**) When compared to paired yoked controls, focal animals showed significant increases in cumulative sleep across two cycles (post 1: Wilcoxon's $W=4088$, $p=0.01$; post 2: Wilcoxon's $W=4009$, $p=0.03$). In the third post-deprivation day, even though the focal flies showed a tendency to have higher cumulative sleep gain, this was not statistically significant (Wilcoxon's $W=3932$, $p=0.06$). All comparisons are made using the Wilcoxon rank sum test, and n=82 each for focal and yoked categories. *<0.05 and NS (not significant). (**E**) Compared to total sleep lost, focal flies do not pay back a significant amount of lost sleep, perhaps owing to normalization with unperturbed controls (see Methods). n=82 each for focal and yoked categories, and n=58 for unperturbed controls *<0.05 and NS (not significant).

The online version of this article includes the following figure supplement(s) for figure 3:

**Figure supplement 1.** 24 hour SD generates multi-cycle post SD sleep pressure signatures.

**Figure supplement 2.** 24 hour SD results in altered sleep architecture across multiple cycles.

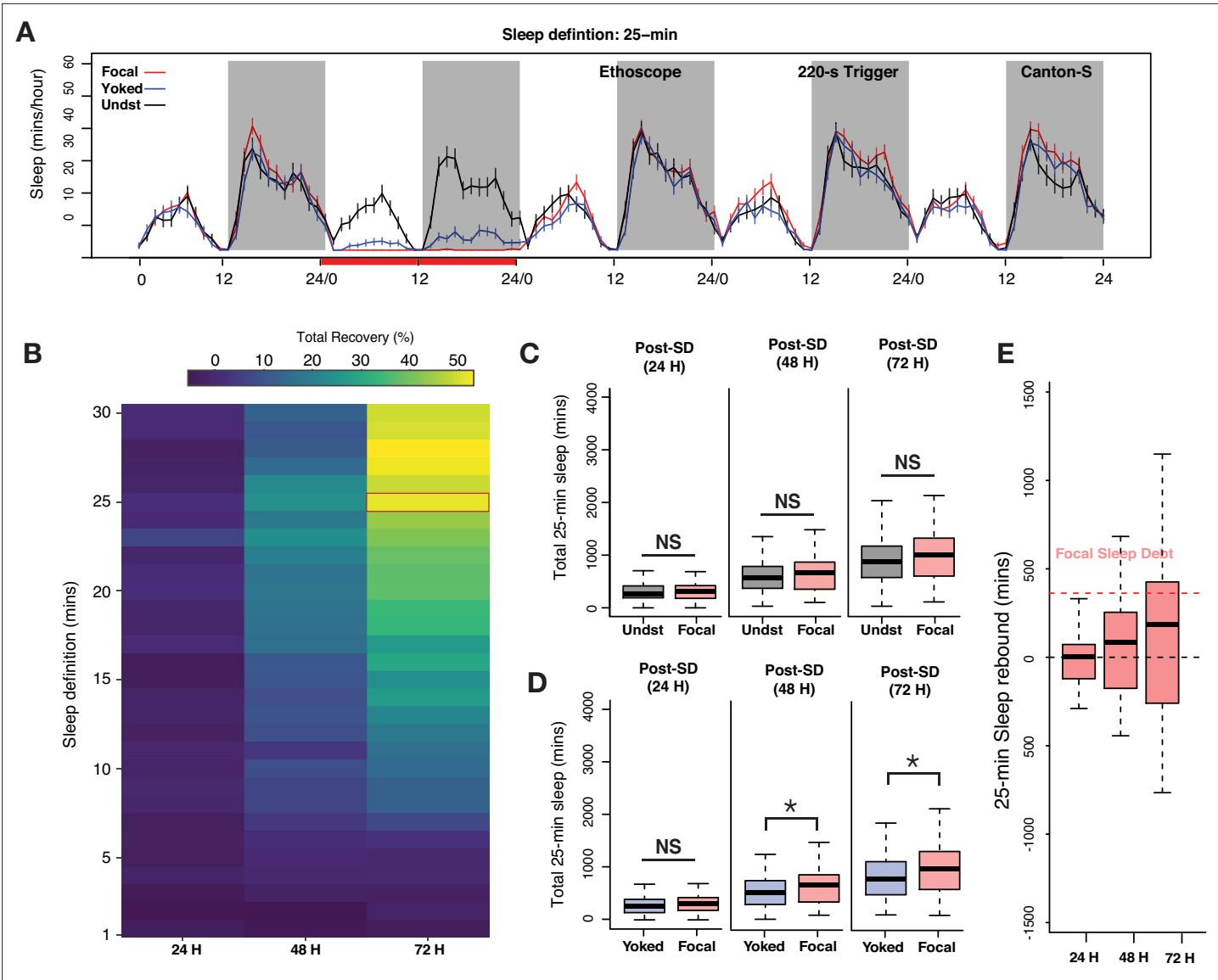

**Figure 4.** Examination of long-bout sleep reveals a multi-cycle sleep rebound that discharges a significant proportion of lost sleep. (**A**) Averaged sleep time-series of long-bout sleep (25 min inactivity criterion) in baseline, deprivation (220 s inactivity-dependent trigger), and three post-deprivation days in focal, yoked, and undisturbed flies. (**B**) Sleep recovery as a function of sleep definition (minutes of inactivity) visualized as a heatmap. When sleep is defined as progressively longer bouts of inactivity, deprived flies paid back a majority of their sleep debt cumulatively over 3 days as the inactivity criterion approaches 25 min. When either the standard sleep definition (5 min or more of inactivity) or active/short sleep criteria (1–5 min) are used, little to no sleep debt is repaid. (**C**) In 3 post-deprivation days, focal flies showed trends of increased cumulative sleep recovery in 25 min sleep but were not significantly different from undisturbed controls (post 1: Wilcoxon's $W$=2458, p=0.73; post 2: Wilcoxon's $W$=2689, p=0.18; post 3: Wilcoxon's $W$=2766, p=0.10). (**D**) Accounting for mechanical stimulation resulted in significant and substantial multi-cycle total sleep increases for focal flies. Interestingly, the first cycle after deprivation did not show significant increases in long-bout sleep in focal flies (post 1: Wilcoxon's $W$=3847, p=0.11; post 2: Wilcoxon's $W$=3987, p=0.03; post 3: Wilcoxon's $W$=4001, p=0.03). (**E**) Normalized sleep rebound for sleep bouts of 25 min or longer across 3 days shows the discharge of sleep debt accrued during the 24 hr of deprivation. Median recovery was more than 50%, while 25% of flies overshoot total sleep loss, thereby showing hallmarks of homeostatic control. n=82 each for focal and yoked categories, and n=58 for unperturbed controls *<0.05 and NS (not significant).

The online version of this article includes the following figure supplement(s) for figure 4:

**Figure supplement 1.** 24 hour SD causes multi cycle post SD increases in long bouts of sleep.

sleep by the third day of recovery, a much higher proportion than was apparent using the currently employed unitary definition of sleep as bouts of inactivity lasting 5 min or more.

Based on this result, we re-examined our 24 hr sleep deprivation data using 25 min of inactivity as our criterion for sleep, thereby focusing on the longer bouts of sleep that likely correspond to a deep sleep state and omitting shorter, presumably shallower, bouts of inactivity from our analysis. Five-day long-bout sleep time-series for focal, yoked, and undisturbed flies are shown in *Figure 4A*. Unperturbed controls displayed two daily peaks of long-bout sleep, with approximately twice as much deep sleep taking place at night, as previously described (*Hendricks et al., 2000*; *Stahl et al., 2017*). Focal flies were, as expected from 220 s inactivity triggers, completely deprived of sleep bouts consisting of 25 min or more. Yoked flies also, as expected, displayed significantly reduced levels of such long-bout sleep compared to unperturbed controls, but such sleep was in fact attained by these controls, though at levels significantly reduced from those displayed by unperturbed controls (*Figure 4A*).

When total long-bout sleep across three successive cycles was quantified, focal flies displayed significant increases in such sleep when compared only to yoked controls on days 2 and 3 of recovery (*Figure 4C and D*). The absence of a difference with yoked flies on day 1 is likely a product of the substantial reduction in long-bout sleep that yoked flies experienced during the deprivation period. Nevertheless, focal and yoked flies revealed differential long-bout sleep rebounds across 3 days of recovery. Remarkably, this rebound was not seen when focal flies were compared to unperturbed controls (*Figure 4C*). Once again, this was likely due to the sleep-pressure-independent effects of mechanical disturbance. To confirm that frequent 25 min bouts of inactivity were not an abnormal occurrence induced by the Ethoscope (i.e., that this state was not induced by the rotations delivered by the Ethoscopes), we confirmed that such long bouts of sleep are frequently observed in unperturbed control flies. Indeed, all unperturbed control flies routinely displayed 25 min or longer bouts of inactivity (*Figure 4—figure supplement 1A*). The sleep architecture of focal flies after sleep deprivation also changed significantly compared to yoked controls where instances of longer and consolidated sleep bouts were significantly increased in the focal flies, although long-bout sleep durations were not significantly different (*Figure 4—figure supplement 1B and C*).

These results suggested that longer bouts of sleep are a more sensitive reflection of homeostatically controlled sleep in flies. This conclusion is supported by the significant recovery of lost long-bout sleep that continues across all 3 days of the observed recovery period (*Figure 4E*). Thus, when the analysis focused on long and presumably deep sleep states, a much larger percentage of such sleep is recovered compared to a consideration of all epochs of inactivity lasting longer than 1 min. At least half of all sleep-deprived flies paid back more than 50% of the sleep they lost during the 24 hr of deprivation, and over 25% of flies overshot their sleep debt, thereby displaying a defining hallmark of homeostatic control (*Figure 4E*). These results lend further support for the idea that long bouts of inactivity represent a sleep state that is distinct from shorter bouts of sleep (*Stahl et al., 2017*; *Tainton-Heap et al., 2021*; *van Alphen et al., 2013*). Furthermore, robust homeostatic recovery of long-bout suggests that our initial failure to detect rebound when 220 s inactivity triggers were used for deprivation (see *Figure 1*) was not simply due to increases in short sleep epochs preventing the build-up of homeostatic sleep drive (*Figure 1—figure supplement 1*), as long-bout sleep displays clear homeostatic rebound even when 220 s triggers allow for substantial levels of short sleep during deprivation.

## Long bout sleep is under potent homeostatic control

In mammals, both REM and NREM stages of sleep are controlled homeostatically, with stage-specific deprivation leading to stage-specific rebound (*Borbély, 1982*; *Borbly and Neuhaus, 1979*; *Endo et al., 1998*; *Endo et al., 1997*). If long-bout sleep indeed reflects a distinct sleep stage in *Drosophila*, selectively depriving flies of it should produce a homeostatic rebound in long-bout sleep, despite the presence of abundant short-bout sleep. To test this prediction, we set the Ethoscope's inactivity-dependent trigger to rotate tubes only when focal flies had been immobile for 22 min as a means to prevent flies from attaining 25 min of uninterrupted inactivity. When using the standard inactivity duration criterion for sleep (5 min or more), this long trigger latency supported what appeared to be largely normal sleep rhythms in both focal and yoked flies (*Figure 5A*). Unsurprisingly, normal sleep patterns were also observed in unperturbed flies (data not shown).

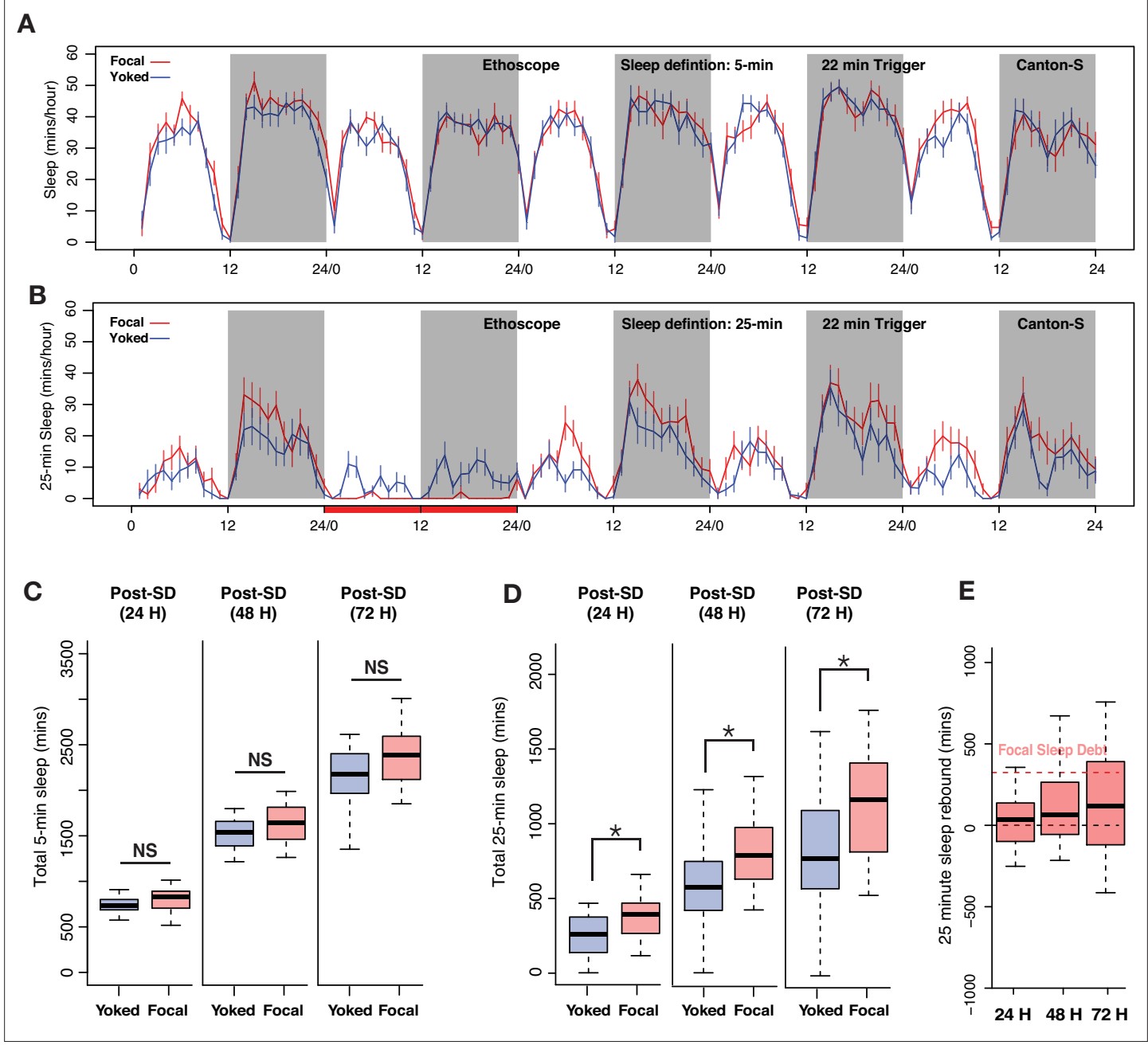

**Figure 5.** Deprivation of only long-bout sleep is sufficient to produce homeostatic responses and elicit multi-day sleep rebound. (**A**) Five-day sleep time courses using the standard inactivity criterion for sleep (5 min or more) in focal-yoked flies during baseline, long-bout sleep deprivation (22 min inactivity triggers), and 3 days of recovery. There were no discernable changes in the average standard sleep measurements between experimental and control flies. (**B**) Five-day sleep time courses of long-bout sleep (inactivity durations of 25 min or more) during baseline, deprivation, and 3 post-deprivation days in focal and yoked flies. (**C**) In the 3 post-deprivation days there were no significant differences in standard sleep between focal and yoked flies (post 1: Wilcoxon's *W*=269, p=0.06; post 2: Wilcoxon's *W*=258, p=0.12; post 3: Wilcoxon's *W*=263, p=0.09). (**D**) Long-bout sleep displayed significant increases in the total amount of sleep over three post-deprivation cycles when compared to yoked controls (post 1: Wilcoxon's *W*=288, p=0.01; post 2: Wilcoxon's *W*=284, p=0.02; post 3: Wilcoxon's *W*=290, p=0.01). (**E**) Long-bout sleep rebound showed over 25% of flies paying back total sleep debt and overshooting in three post SD cycles. Focal sleep debt reflects the total sleep lost during deprivation, which was normalized to sleep amounts in undisturbed controls. n=20 each for focal and yoked categories *<0.05 and NS (not significant).

The online version of this article includes the following figure supplement(s) for figure 5:

**Figure supplement 1.** Time course of control *CS* flies for five LD cycles from a different run (see and compare with **Figure 4**), showing the remarkable consistency between runs in the amount of 25 min sleep.

**Figure supplement 2.** Selective deep sleep deprivation causes multi cycle post SD increases in long bouts of sleep.

Employing a 25 min inactivity duration criterion to remove short-bout sleep from our analysis revealed that 22 min triggers effectively deprived flies of long-duration sleep while permitting yoked flies to achieve appreciable amounts of such sleep during focal deprivation (*Figure 5B*). Unperturbed control flies once again displayed the bimodal long-bout sleep rhythms described above, with substantially more nighttime than daytime deep sleep (*Figure 5—figure supplement 1*). When behavior was analyzed using the standard 5 min inactivity criterion, which would include short-bout sleep, focal flies displayed no significant increases in total sleep when compared to yoked controls (*Figure 5C*). In contrast, when short-bout sleep was eliminated from the analysis through the use of a 25 min inactivity criterion, focal flies displayed significantly higher levels of long-bout sleep across all 3 days following deprivation (*Figure 5D*). Remarkably, the relative levels of post-deprivation sleep displayed by the focal flies appeared to increase over the course of the 3 days of observation, suggesting that homeostatic responses to long-bout sleep deprivation may persist for an extended period of time (*Figure 5D*). This rebound produced a significant recovery of lost sleep (*Figure 5E*), one that may have continued to build over additional sleep/wake cycles. No such patterns were observed when focal flies were compared to unperturbed controls (data not shown), thereby once again highlighting the importance of yoked controls.

Long-bout-specific deprivation also produced significant effects on sleep architecture. Focal flies were characterized by an increased number of bouts that were longer than 25 min compared to yoked controls (*Figure 5—figure supplement 2B*), although the durations of these longer sleep bouts did not increase (*Figure 5—figure supplement 2D*). No differences in the number or duration of long-bout sleep between focal and yoked flies were observed prior to the deprivation cycle (*Figure 5—figure supplement 2A and C*). Our results suggest that long-bout sleep, previously shown by others to represent a relatively deep sleep state, is under strong homeostatic control and highlights the importance of controlling for the sleep-independent effects of mechanical sleep deprivation on behavior.

## Yoked controls are necessary to differentiate effects of sleep deprivation from effects driven by mechanical perturbation

A major goal of sleep science is to identify the molecular, cellular, and physiological processes mediating the rise and fall of sleep pressure and how these changes operate in the brain to promote sleep or wakefulness. In the fly, previous work by others has described cellular (*Blum et al., 2021*; *Donlea et al., 2014*; *Liu et al., 2016*) and physiological (*Pimentel et al., 2016*; *Tainton-Heap et al., 2021*) correlates of sleep pressure, based on the effects of mechanical sleep deprivation. Given our behavioral results indicating significant sleep-independent effects of mechanical deprivation that appear to mask homeostatic sleep responses, we sought to examine the potential utility of yoking for differentiating sleep-pressure-driven changes from the sleep-independent effects of mechanical perturbation. Toward this end we conducted a simple molecular screen in the fly brain using matrix-assisted laser desorption/ionization-time-of-flight (MALDI-TOF) mass spectrometry (MS) (*Veerasammy et al., 2020*). This method supports the MS analysis of fly head sections (*Figure 6A*; *Salisbury et al., 2013*) and consists of three major steps: the crystallization of a matrix and analytes within the tissue section, the laser ionization of analytes within this matrix, and TOF MS to analyze the molecules present (*Veerasammy et al., 2020*). This method provides sufficient spatial resolution to detect the presence of molecules defined by their weight over charge ratios (m/z), specifically within the central brain (*Figure 6B*). Using our behavioral sleep deprivation approach (*Figure 2E–G*), we examined the molecular responses within the central brains of focally deprived flies, paired yoked controls, and unperturbed flies following 24 hr of sleep deprivation using 220 s inactivity triggers.

Using 2,5-dihydrobenzoic acid (DHB) as a positive-ion matrix, which is considered useful for 'universal analysis' and the detection of diverse molecular types (*Snovida et al., 2006*), we sampled m/z value peaks of up to 1300 and detected 188 distinct peaks in the central brain regions of *Drosophila* head sections. Each m/z ratio peak measurement was made specifically for the central brain by registering the MALDI-TOF signal directly over the histological image of our tissue sections (dashed square, *Figure 6B*). When we compared the heights of these peaks between focal, sleep-deprived flies and unperturbed controls, we found 127 peaks that displayed statistically significantly higher amplitudes in sections from focal flies compared to those of unperturbed flies (*Figure 6C*). Thus, when compared only to unperturbed flies, sleep deprivation in focal flies was accompanied by increases in a large

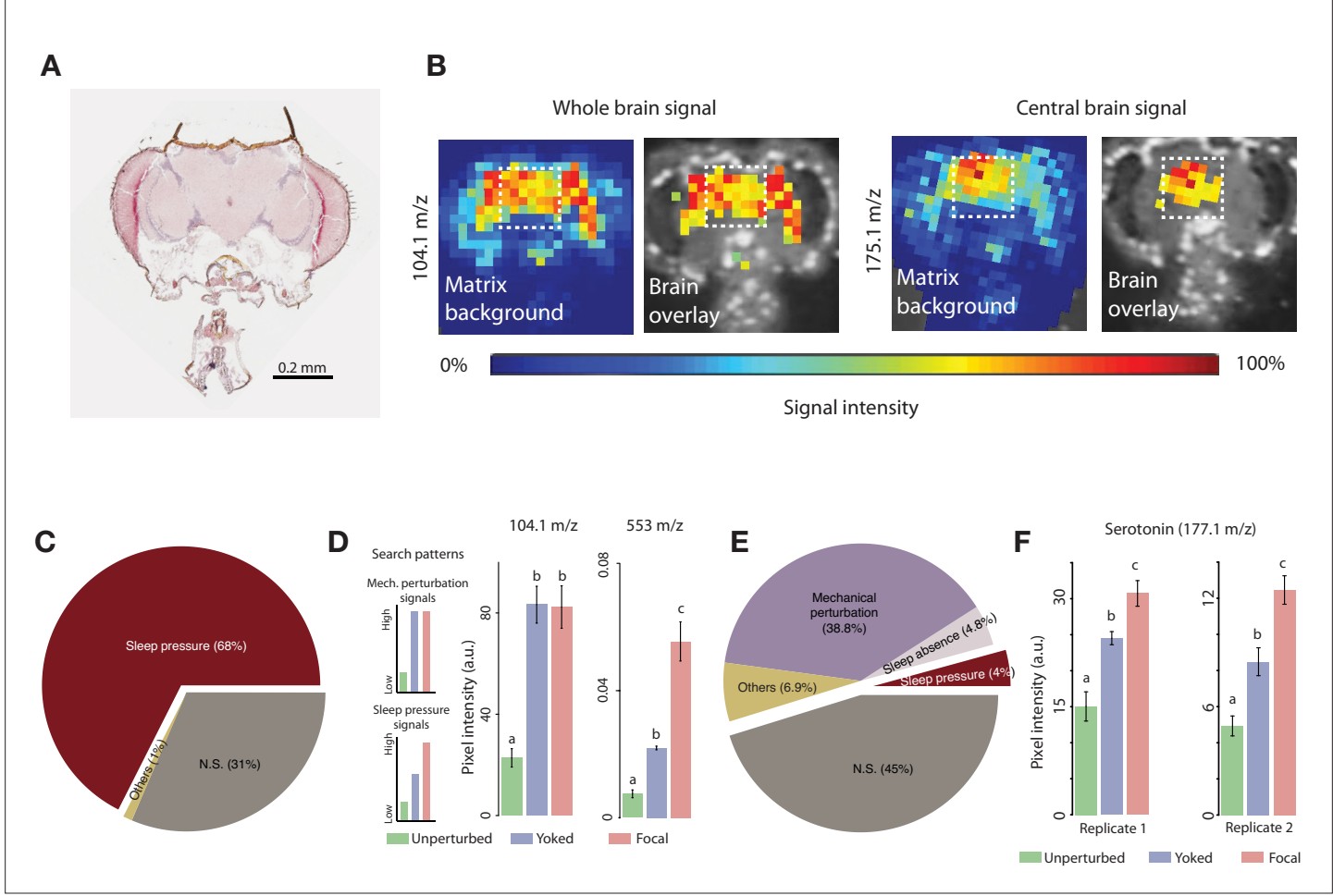

**Figure 6.** Identifying sleep substances using matrix-assisted laser desorption/ionization-time-of-flight mass spectrometry (MALDI-TOF MS). (**A**) Representative hematoxylin-eosin stained cryosection of a fly head section. (**B**) Distribution of two representative molecules (104.1 m/z and 175.1 m/z) within a single head section. Note the near-zero relative intensity outside the brain region. Images are overlays of the high-intensity signals on the cryosection used to detect the signal and highlights brain-specific distribution of these molecules. The white dotted box represents the area, corresponding to the central brain, from which signal intensity was quantified. (**C**) Post-deprivation increases in the relative abundance of molecules in focal (deprived) flies when mechanical perturbations are not accounted for yields 127 positive hits (out of 188 molecules – 68%) in the first replicate. We also found 59 peaks that were not significantly different between focal and undisturbed flies, and 2 peaks that were significantly different, but lower in the focal than in undisturbed flies, which we termed as 'Others'. (**D**) Search patterns of molecules tracking mechanical perturbation signals (top) and sleep pressure-specific signals (bottom). Example peaks (m/z 104.1 and m/z 553) showing each of the two categories. Values plotted are mean ± SEM. (**E**) Pie chart representing the number of molecules in various categories when yoking is employed. The number of molecules reflecting sleep pressure was eight (4%) for the first replicate. We identified an additional category of molecules that may track sleep and refer to it as 'Sleep Absence'. Molecules that track the complete absence of sleep (as in the case of focal flies, but not in partial sleep loss situations as in the yoked flies) are classified under this category, and we found nine such candidates. These nine molecules had indistinguishable values between yoked and undisturbed flies which were both significantly lower than focal flies. The number of sections from independent brains that were used are as follows: focal = 6; yoked = 3; unperturbed = 4. (**F**) One of the detected sleep-tracking molecules was serotonin, detected at 177.1 m/z, which showed significant sleep pressure tracking changes in both replicate runs. The number of sections from independent brains that were used for the second replicate are as follows: focal = 7; yoked = 4; unperturbed = 4 (replicate 1: *Kruskal-Wallis critical value* = 9.32, df = 2, p=0.009; replicate 2: *Kruskal-Wallis critical value* = 11.56, df = 2, p=0.003). Bars with the same letters are not significantly different from each other.

The online version of this article includes the following figure supplement(s) for figure 6:

**Figure supplement 1.** Spotted amines in matrix-assisted laser desorption/ionization-time-of-flight (MALDI-TOF) mass spectrometry (MS) experiments and detected serotonin concentration at m/z 177.07.

proportion of the molecules examined. However, it is highly likely that the mechanical stimulation associated with sleep deprivation causes many of these molecular changes within the brain and such increases might not track sleep pressure. We therefore asked the extent to which controlling for mechanical insult using yoked controls might allow us to differentiate increases explained by sleep pressure from those that were simply produced by mechanical stimulation.

If a molecule's abundance is increased simply as consequence of mechanical insult during sleep deprivation, we would expect head sections from both focal and yoked flies to display similar concentrations of that molecule that are elevated compared to unperturbed controls (*Figure 6D*, left-top). In contrast, if a molecule's abundance is increased specifically as consequence of sleep pressure, we predict that it will be present at relatively low levels in sections from unperturbed controls, significantly higher levels in sections from focal flies, and with intermediate levels in yoked controls that are statistically distinguishable from the other two conditions (*Figure 6D*, left-bottom). Example m/z peaks that show these characteristic patterns in the central brain are shown in *Figure 6D* (right; m/z values of 104.1 and 553). When yoked controls are included in the analysis, approximately half of the discernable m/z value peaks displayed statistically indistinguishable levels in the central brain across the undisturbed, yoked, and focal samples (*Figure 6E*, 'NS', 45%). Seventy-three of the m/z peaks displayed the relative abundances expected of molecules tracking mechanical perturbation (38.8%), whereas only eight displayed the relative abundances expected of molecules tracking sleep pressure (~4%, *Figure 6E*). Thus, the standard method of comparing sleep-deprived flies to unperturbed controls produced 127 candidate sleep-tracking molecules, whereas the inclusion of yoked controls produced only eight candidates. Thus, many of the molecular differences between sleep-deprived flies and unperturbed controls can likely be explained as responses to mechanical stimulation rather than to the loss of sleep.

We replicated this molecular screen in an independent MALDI-TOF experiment. This run detected slightly fewer m/z peaks (170 compared to 188 in the first replicate). Despite the markedly lower sensitivity of this replicate, we found that two of the eight molecules identified in the first screen as tracking sleep pressure, once again displayed the relative abundances expected of a sleep pressure tracking molecule. One of these had an m/z peak at ~553, and the other at 177.1 m/z in the DHB matrix. The 177.1 m/z peak has previously been shown to correspond to serotonin (*Cao et al., 2019*; *Moriarty et al., 2011*) and fits the molecular weight range of serotonin (176.2) in a positive ion matrix like DHB, which adds a cation with a single positive charge to the m/z peak, which for serotonin would result in a peak at ~177 m/z. Furthermore, we confirmed that this m/z value reflected serotonin by spotting serotonin on slides imaged by MALDI-TOF MS and confirmed strong peaks at m/z=177 for regions of interest corresponding to these spots (*Figure 6—figure supplement 1*). Thus, serotonin would appear to be a strong candidate for a molecule whose abundance reflects sleep pressure in the fly's central brain (see Discussion), that follows the expected pattern across unperturbed, yoked, and focal flies (*Figure 6F*). An unknown molecule with an m/z value of ~553 is likewise also a strong candidate for another such molecule. Although it is impossible to guess the exact nature of this molecule without further extensive experimentation, the m/z ratio indicates that this molecule is larger than amino acids, other biogenic amines, or other neurotransmitters (for instance, dopamine, octopamine, and GABA), and smaller than large peptides, or peptide receptors.

## Discussion
### The impact of the sleep-independent effects of mechanical sleep deprivation on the measurement of homeostatic sleep rebound

In order to study the sleep homeostat, one must prevent sleep through the imposition of prolonged wakefulness (*Bentivoglio and Grassi-Zucconi, 1997*; *Henri, 1895*; *Patrick and Gilbert, 1896*; *Rechtschaffen et al., 1983*). However, the very act of sleep disruption is expected to cause not just the build-up of sleep pressure, but also other potentially confounding responses to the stimulus used to prevent sleep. The specific confounds attending sleep deprivation will depend on the method, duration, and time of day of the sleep-depriving stimulus employed, and all forms of experimental sleep deprivation are expected to produce such confounds, including methods based on chemo-, thermo-, or opto-genetics. Any sleep deprivation method that produces increased locomotion will increase muscular activity, which will be attended by sleep-independent effects. Chronic mechanical

agitation also produces stress (*Meerlo et al., 2008*; *Sgoifo et al., 2006*; *Tartar et al., 2009*), and even gentle handling of rodent subjects, a means of deprivation thought to cause little stress, has significant effects on feeding behavior following deprivation (*Dukanovic et al., 2022*). Such confounding, sleep-independent effects can mask homeostatic sleep responses and must therefore be accounted for whenever possible. Here, we have attempted to account for the sleep-independent effects of mechanical sleep deprivation in *Drosophila,* the most commonly used method in the field of fly sleep research. Sleep-independent effects of any method of deprivation are likely multi-factorial. The utility of the yoked control paradigm introduced by *Rechtschaffen et al., 1983*, is that these effects do not need to be completely understood in order to be accounted for. Though it is not within the scope of our study to completely account for the sleep-independent effects of mechanical deprivation, we suggest a partial explanation: the depriving stimuli itself cause increases in locomotor activity in deprived flies that likely antagonize the increased drive to sleep in deprived flies (*Figure 2—figure supplement 1*).

To assess the degree to which the frequency of mechanical agitation affects subsequent sleep rebound in *Drosophila*, we examined flies that were sleep deprived for 24 hr with different frequencies of mechanical perturbation. The less frequent agitation produced significantly less sleep rebound, despite having produced similar amounts of sleep loss. Thus, a sleep-pressure-independent effect of mechanical perturbation appeared to be the primary determinant of post-deprivation sleep. An alternative explanation for this result was that short bouts of inactivity may have provided brief but restorative sleep that the longer perturbation frequencies were not effective at depriving. Indeed, 220 s inactivity triggers produced increases in brief (between 1 and 5 min) periods of inactivity during deprivation (*Figure 1—figure supplement 1*). Such short sleep epochs have been proposed to be a distinct sleep state from the standard sleep traditionally measured in flies (*Anthoney et al., 2023*; *Tainton-Heap et al., 2021*). However, based on our estimates of p(Doze), a metric of sleep pressure (*Figure 3—figure supplement 1*), which is persistently high for three cycles post sleep deprivation, we do not think that this short sleep epoch prevented the build-up of homeostatic sleep drive, at least in this context. Furthermore, based on a large body of work conducted on humans and other mammals, we would not necessarily expect increases in one sleep state to prevent the homeostatic regulation of another.

Though it is possible that increases in the amount of brief short sleep bouts may have reduced the build-up of sleep pressure during deprivation experiments employing 220 s inactivity triggers, our results strongly support the conclusion that mechanical stimulation produces sleep-independent effects that can mask homeostatic sleep rebound. Six, 12-, and 24 hr of nighttime sleep deprivation using frequent (20 s) triggers failed to produce the expected relationship between the amount of sleep lost and the magnitude of rebound during the recovery day (*Figure 1J*). The failure of rebound to track the amount of sleep loss indicates that the comparison of sleep-deprived flies and unperturbed controls is not sufficiently sensitive to detect the differential sleep rebounds expected from flies that have suffered significantly different levels of sleep loss (*Figure 1*).

A previous attempt to control for the sleep-independent effects of mechanical perturbation delivered mechanical stimuli matched to sleep-deprived flies, but delivered during periods of high activity (i.e., wakefulness), thereby exposing the control flies to mechanical shaking while resulting in negligible amounts of sleep loss (*Harbison and Sehgal, 2009*). However, mechanical shaking only during times of wakefulness increased molecular signatures of stress, even when given for relatively short periods (*Harbison and Sehgal, 2009*). To minimize the extent of mechanical perturbation necessary to prevent sleep and to control for the sleep-independent effects of such perturbation, we adopted the Ethoscope, a video-based system that tracks single flies and delivers a mechanical stimulus, the rapid rotation of the glass tube containing the fly, only when a fly meets a user-defined inactivity duration criterion (*Geissmann et al., 2017*). Thus, rather than agitating all the flies of the sleep-deprived condition at a given frequency throughout the entire deprivation period, the Ethoscope prevents single flies from attaining specific durations of inactivity, just before meeting a user-defined duration. Previous work using this system has revealed remarkably low levels of homeostatic rebound and no sleep deprivation-associated death (*Geissmann et al., 2019*).

Though the Ethoscope produces sleep deprivation using fewer mechanical perturbations, it nevertheless delivers significant mechanical perturbation whose sleep-independent effects might mask homeostatic sleep responses (*Figure 2A–D*). For this reason, we introduced yoked controls to the

Ethoscope platform, allowing us for the first time to disentangle effects associated with sleep pressure from those best explained by the mechanical insult delivered during sleep deprivation (*Figure 2E–G*). When flies were prevented from attaining 5 min bouts of inactivity – the standard criteria for a sleep-like state in the fly – using 220 s inactivity triggers in Ethoscopes, they failed to show significant homeostatic rebound when their sleep was compared to unperturbed controls. It is noteworthy here that *Geissmann et al., 2019*, used a 220 s trigger and reported small but significant rebound compared to undisturbed controls. We believe this discrepancy is based on two differences between our studies. First, the rebound analysis window was only 3 hr (ZT0–03) in Geissmann et al. but was for a full 24 hr cycle in our study. Second within Geissman et al.'s rebound analysis window, a regression model was used to derive expected sleep amounts (based on pre-deprivation trends), whereas we compared raw data from undisturbed controls. It is possible that small rebounds may have been detected if Geissman's methods had been used in our study. In contrast to small or non-existent rebound when comparing deprived and unperturbed flies, the comparison of these deprived flies to their yoked controls revealed a significant, albeit small, increase in sleep following deprivation (*Figure 2F and G*). Thus, yoking revealed homeostatic rebound that was not apparent when deprived flies were compared to controls that had not experienced mechanical stimulation. However, when compared to yoked controls, the homeostatic response of sleep-deprived flies was quite small, amounting to very little recovery of lost sleep (*Figure 3E*).

## Sleep as a non-unitary state in *Drosophila*: long-bout sleep is a distinctly regulated sleep stage

In mammals, sleep is not a unitary state but rather consists of distinguishable sleep stages (*Davis et al., 1937*; *Loomis et al., 1937*). For example, REM and NREM sleep represent distinct physiological states that differ in their relationship to sleep pressure and the circadian system (*Borbély, 1982*; *Borbly and Neuhaus, 1979*; *Czeisler et al., 1980*; *Endo et al., 1997*). Slow-wave NREM sleep appears to be the mammalian sleep state most strongly controlled by the daily homeostatic control of sleep: initial sleep cycles following prolonged wakefulness are rich in slow-wave sleep, and this form of sleep is also prevalent following sleep deprivation (*Berger and Oswald, 1962*; *Ursin, 1971*). For the majority of studies using *Drosophila*, sleep has been treated as unitary state, with all inactivity bouts of five minutes or more being treated as sleep. However, a growing body of works suggests the existence of distinct sleep stages in the fly. For example, in one of the very first papers describing sleep-like states in *Drosophila*, Hendricks et al. reported differences in durations of sleep bouts consisting of epochs of inactivity lasting 5 min or more (*Hendricks et al., 2000*). The longest bout lengths occurred at the beginning of the subjective night under constant conditions between circadian time 13.5 and 16. More recently, quantitative approaches using hidden Markov models have suggested the existence of light and deep sleep states in the fly (*Wiggin et al., 2020*; *Xu et al., 2021*).

Flies display both longer and deeper bouts of sleep at night (*Hendricks et al., 2000*; *van Alphen et al., 2013*) and deeper sleep is associated with distinct patterns of global brain activity when compared to brain activity during wakefulness or during more brief and shallow bouts sleep (*Nitz et al., 2002*; *Tainton-Heap et al., 2021*). Longer sleep bouts are also attended by lower metabolic rates, where bout lengths of 60 min were shown to be associated with the lowest rates (*Stahl et al., 2017*), a phenomenon similar to that observed for deeper NREM sleep stages in humans (*Berger and Phillips, 1995*). Finally, flies display a stereotyped pattern of proboscis movements that appear to drive waste clearance (*van Alphen et al., 2021*), reminiscent of the changes in fluid dynamics observed in the cerebrospinal fluid of sleeping mammals, that occur at times corresponding to deep sleep stages (*Fultz et al., 2019*). The time of peak occurrence of these proboscis movements were characterized by a significant reduction of brain wave activity compared to sleep at other times (*van Alphen et al., 2021*).

Physiological and metabolic signatures of deep sleep in the fly both suggest that they correspond to sleep bouts that are significantly longer than the standard sleep definition of 5 min of inactivity and multiple lines of evidence suggest that deeper sleep states are likely reached upon attaining 15–30 min of inactivity (*Tainton-Heap et al., 2021*; *van Alphen et al., 2013*). Finally, recent work from our lab has suggested that bouts of inactivity longer than anything between 30 and 60 min are stronger reflections of the daily homeostatic control sleep than shorter bouts of sleep (*Abhilash and Shafer, 2023*). How might such long bouts of inactivity relate to homeostatic sleep rebound?

Homeostatic sleep rebound, as traditionally measured in flies, appears to differ from mammalian sleep rebound, in that it is relatively modest in magnitude and appears to largely run to completion during the first few hours following the offset of deprivation (*Huber et al., 2004*). In contrast, mammalian rebound appears to play out over several sleep-wake cycles (*Coborn al., 2019*; *Mistlberger et al., 1983*; *Nakazawa et al., 1978*; *Rechtschaffen et al., 1999*). Remarkably, when we biased our analysis to bouts of inactivity lasting 25 min or more, clear and prolonged homeostatic rebound of sleep was observed following deprivation. This rebound lasted several days (*Figure 4*) and discharged a large proportion (>50%) of the long-bout sleep debt suffered during deprivation (compare *Figures 3E–4E*). This result further supports the notion that long durations of inactivity more sensitively reflect the action of the sleep homeostat.

When we deprived flies only of inactivity bouts that were longer than 22 min (see Methods), a duration that we chose based on its effectiveness at reliably preventing 25 min epochs of inactivity, we observed a homeostatic increase in long-bout sleep, while seeing no such homeostatic response in standard sleep (*Figure 5*). Moreover, the homeostatic rebound of long-bout sleep was apparent on the very first day of recovery (*Figure 5B and D*) as opposed to the rebound seen following deprivation using shorter inactivity triggers (*Figure 4A and D*). This is probably due to the fact that long-bout sleep deprivation engaged mechanical triggers only 12–14 times on average over the 24 hr of deprivation, and thereby likely reduced significantly the sleep-independent effects of mechanical perturbation.

## Yoked controls are critical for the examination of molecular and physiological correlates of sleep pressure

A major goal of sleep science is to identify the molecular, cellular, and physiological mechanisms underlying the build-up of sleep pressure and how such pressure promotes sleep. As has been recognized since the founding of sleep science (*Kleitman, 1923*; *Kleitman, 1939*), inescapable confounds attend sleep deprivation experiments. The increased activity caused by most forms of experimental sleep deprivation is likely to have myriad behavioral, physiological, and metabolic effects. Mechanical sleep deprivation through vortexing, the most commonly used method of sleep deprivation in *Drosophila* research, is particularly problematic in this regard, as it appears to cause significant physical stress (*Harbison and Sehgal, 2009*).

The sleep-independent effects of deprivation protocols represent a significant confound and pose a major challenge to the identification of the molecular and physiological correlates of sleep pressure. If a molecular or physiological response to deprivation truly represents an increase in sleep pressure, it must be established that it is not simply a response to the stimuli used to prevent sleep. A fundamental challenge for the sleep scientist working in the fly system therefore is to differentiate between the effects of increased sleep pressure from the sleep-independent effects of the stimuli used to prevent sleep. The use of yoked controls and longer duration inactivity triggers in our study appears to be a useful means of accounting for the sleep-independent effects of mechanical sleep disruption on behavior (*Figures 2–5*). We therefore wondered if this approach would allow us to differentiate molecular changes that were specific to sleep pressure from changes produced directly by the effects of mechanical perturbation.

Given the extent of disturbance necessary to keep flies awake during prolonged sleep deprivation, we predicted that a large number of molecules whose abundance increases in response to sleep deprivation would reflect responses to mechanical stimulation rather than sleep pressure driven increase. Indeed, when we compared the brains of sleep-deprived files to unperturbed controls, around 70% of all detected molecules showed a significant increase in abundance. However, when we employed yoked controls as an additional control to identify which molecular weights were higher in deprived flies compared to controls that had experienced identical mechanical perturbation but significantly less sleep deprivation, we found that only approximately 4% of detectable molecules reflected sleep pressure (*Figure 6*). Thus, many of the molecular changes in response to sleep deprivation can likely be explained by the effects of mechanical disturbance and not by sleep pressure. This result highlights the importance of yoked controls for confirming that specific molecules, genes, and cells mediate sleep homeostasis rather than the behavioral or physiological responses to prolonged mechanical stimulation. Such controls will be critical for the vetting of physiological or cellular correlates of sleep pressure. More specifically, the limited molecular screen we describe here supports the feasibility of

using yoked controls to discover sleep substances within the fly brain, molecules whose abundance rises and falls with sleep pressure that are used by sleep control centers to exert homeostatic control of sleep (*Borbély, 1986*).

## Serotonin is a candidate sleep substance in the *Drosophila* central brain

The inclusion of yoked controls in our experiments revealed that very few of the molecular changes produced by mechanical sleep deprivation tracked sleep loss. One molecule that reliably tracked sleep pressure in independent experiments was serotonin, which, based on our results, can be considered a strong candidate sleep substance in the fly's central brain. A previous study using HPLC to examine serotonin levels in dissected brains failed to observe serotonin increases in sleep-deprived flies (*Davla et al., 2020*). However, the spatial resolution provided by MALDI-TOF imaging suggests that sleep loss-driven serotonin increases take place in the central brain and not the optic lobes (e.g., *Figure 6B and F*). We therefore suspect that serotonin does not build up throughout the optic lobes and that whole-brain assays might therefore fail to detect the sleep loss-induced changes we describe here. The identification of serotonin as a candidate sleep substance is notable, as convergent evidence in the fly strongly supports the conclusions that serotonin signaling promotes sleep in *Drosophila*.

Pharmacological increases in serotonin and the excitation of serotonergic neurons produce increases in baseline sleep in the fly (*Haynes et al., 2015*; *Pooryasin and Fiala, 2015*; *Yuan et al., 2006*). Similarly, blocking synaptic transmission in single pairs of serotonergic neurons have been reported to reduce baseline sleep (*Alekseyenko et al., 2014*). Genetic loss of serotonin synthesis decreases baseline sleep, as does the loss of 5HT-R-1A and 5HT-R-2B serotonin receptors (*Qian et al., 2017*; *Yuan et al., 2006*). Loss of serotonin and the serotonin receptor 5HT-R-2B also results in a reduction in sleep rebound following deprivation (*Qian et al., 2017*). Furthermore, the genetic loss of arylalkylamine *N*-acetyltransferase, which normally functions to inactivate serotonin and dopamine, is accompanied by both increased homeostatic sleep rebound and brain serotonin levels following sleep deprivation, consistent with serotonin acting as a sleep substance during mechanical deprivation (*Davla et al., 2020*). Finally, loss of the serotonin transporter dSERT, which normally functions to remove extracellular serotonin, is associated with significant increases in sleep (*Knapp et al., 2022*). Thus, convergent evidence supports the conclusion that serotonin promotes both basal sleep and homeostatic increases in sleep following deprivation.

According to the two-process model of sleep regulation, basal seep and homeostatic increases in sleep are driven by 'process S', which is thought to be mediated by somnogens, molecules that rise and fall with sleep pressure (*Borbély, 1986*; *Borbély, 1982*; *Borbély and Achermann, 1999*). Our MALDI results, in conjunction with previous work establishing a sleep-promoting role for serotonin in the fly, suggest that serotonin levels build in the central brain during sleep deprivation and increase the pressure to sleep through actions of serotonin receptors. Remarkably, this very model was proposed by Jouvet to reconcile conflicting results regarding serotonin's role in the control of mammalian sleep (*Jouvet, 1999*). A significant body of work has established that increasing serotonin in the mammalian brain produces sleep increases and that the loss of serotonin results in insomnia. These observations were the basis of the serotonergic theory of sleep (*Jouvet, 1999*). Serotonergic neurons were subsequently shown to be active during wakefulness and this was taken as evidence against the hypothesis that serotonin promotes sleep (*Jouvet, 1999*). However, neurons producing a sleep substance would be expected to be active during wakefulness and to thereby promote sleep pressure during prolonged bouts of wakefulness. Serotonin neurons were hypothesized to act in this way by Jouvet in order to reconcile the wake-active nature of serotonin neurons with the serotonergic theory of sleep (*Jouvet, 1999*). Our work strongly implicates serotonergic signaling as a component of process S in the *Drosophila* brain and sets the stage for future work to test the predictions of the serotonergic theory of sleep in the fly.

## Conclusion

The work described here provides both new insights into sleep homeostasis in flies and new methodologies that are likely to improve our ability to detect homeostatic sleep responses and identify molecular correlates of sleep pressure in this species. Our work indicates that the mechanical perturbation most commonly used to deprive flies of sleep produces both behavioral and molecular changes that potentially mask the sleep-pressure-specific changes that underly homeostatic sleep control. The

results of our work also reveal that long bouts of sleep, which have been shown by several studies to represent a deep sleep state (*Tainton-Heap et al., 2021*; *van Alphen et al., 2013*), are a more sensitive indicator of sleep pressure and rebound than the previously used unitary definition of sleep as 5 min or more of inactivity. Remarkably, when analysis is focused on longer (~30 min) bouts of inactivity, homeostatic control of sleep in the fly appears much more similar to that seen in mammals, in that a larger proportion of sleep debt is repaid and is discharged over multiple sleep-wake cycles. We predict that this larger magnitude of sleep rebound will be of great utility to the field, increasing the dynamic range over which the effects of molecular and physiological alterations can be examined and thereby increasing the likelihood of discovering new molecular and cellular components of sleep homeostasis.

## Methods

### Fly stocks and husbandry

Flies were reared on Corn Syrup/Soy media made by Archon Scientific (Durham, NC, USA) under a 12 hr:12 hr light:dark (LD) cycle at 25°C and 60–70% humidity. Male wild-type *Canton-S* (*CS*; BDSC stock number: 64349) and $w^{1118}$ (BDSC stock number: 3605) flies were used for the experiments.

### Sleep assay and analysis

One- to three-day old male flies were collected in groups of 30 into Corn Syrup/Soy containing vials and were subsequently isolated under $CO_2$ anesthesia and loaded into glass tubes (70 mm × 5 mm × 3 mm [length × external diameter × internal diameter]) containing 5% sucrose and 2% agar when they were 5–7 days of age. The loading was done at least 24 hr prior to the beginning of behavioral experiments. Flies were allowed to adjust to the tubes for the remainder of the loading day and no data from this day was subject to analysis, which began using data from the next LD cycle. Locomotor activity was measured using two independent methodologies: DAMs (TriKinetics, Waltham, MA, USA), and Ethoscopes built in our lab based on resources provided by the Gilestro Lab (Imperial College London) (*Geissmann et al., 2017*).

The DAM system employs infrared sensors for each tube and records beam crossing at the tube's mid-point as a measure of locomotor activity. Beam crossings were recorded every 20 s and from these data, inactivity bouts of 5 min or more were used as the standard definition of sleep (*Hendricks et al., 2000*; *Shaw et al., 2000*). For Ethoscope experiments, continuous video tracking via infrared cameras were done for individual flies using Raspberry Pi (https://github.com/raspberrypi/documentation/, copy archived at *Allan, 2023*). All TriKinetics data were analyzed using codes from the *phase* package on R (*Abhilash, 2023*). Ethoscopes quantified maximal velocity in mm/s in 10 s epochs. The maximal velocity data was converted to sleep data using custom R scripts (https://github.com/abhilashlakshman/ethoscopeCodes_eLife, copy archived at *Lakshman, 2023*) and downstream analyses were carried out using modified codes from the *phase* package (*Abhilash, 2023*). Maximal velocities of <1 mm/s for any epoch were considered an instance of immobility (*Geissmann et al., 2017*), and sleep was then computed using the standard 5 min inactivity criterion, with immobility lasting 300 s (30 epochs) or more considered as a bout of sleep. This criterion was subsequently adjusted to bias our analysis toward longer bouts of sleep.

Homeostatic sleep rebound analysis: Total sleep displayed by sleep-deprived flies on days 1, 2, and 3 following sleep deprivation (see below) were calculated across each 24 hr day, subtracted from the baseline sleep displayed on the day before deprivation. Normalization was carried out using the following formula ([$Focal_{PostDeprivationSleep}$ – $Focal_{PreDeprivationSleep}$] – [$Unperturbed_{PostDeprivationSleep}$ – $Unperturbed_{PreDeprivationSleep}$]). Sleep debt accrued in sleep-deprived flies was quantified by subtracting baseline 24 hr sleep on the day before deprivation from total sleep measured during the 24 hr deprivation window.

### Mechanical sleep deprivation

24 hr mechanical deprivation was carried under 12/12 L:D cycles with mechanical deprivation starting at ZT00 (lights-on) following 1 day of baseline sleep measurement. 12 hr deprivation commenced at lights-off (ZT12) following 1 day of baseline sleep measurement. Six hr deprivation commenced at

ZT18 (6 hr after lights-off) following 1 day of baseline sleep measurement. The temperature was kept constant at 25°C with 60–70% humidity.

## Mechanical sleep deprivation using DAM and mechanical shakers

Vortexers (Fisherbrand Analog MultiTube Vortexer, Catalog # 02-215-450) were used as previously described to sleep deprive animals in DAMs (*Kayser et al., 2015*). DAMs housing 32 flies within glass tubes were shaken randomly for 2 s every 20 s, 120 s, or 220 s across the entire period of deprivation. The MultiTube Vortexer was set at a shaking intensity of four.

## Inactivity-dependent sleep deprivation using Ethoscopes

Using the rotational module, we sleep deprived flies with a 220 s inactivity threshold. DAM tubes were rotated at ~420 rpm for 1 s every time an individual fly was inactive for 220 s.

## Yoked-controlled mechanical deprivation in the Ethoscope

The Ethoscope rotational module (*Geissmann et al., 2017*) was used to develop a yoked-controlled mechanical deprivation platform. The Ethoscope has been used previously to rotate single tubes only when the animal has been inactive for specific durations of time (as opposed to shaking randomly and without regard to behavioral state as is the case of vortexing DAMs). We altered the closed-loop feedback control (see below) in the Ethoscope to rotate two tubes at the same time based on the inactivity state of only one of the two flies. The fly whose behavior determined the rotation of the tubes is referred to here as the focal fly, whereas the paired fly that received identical time-matched rotations without regard to its behavioral state is called the yoked fly. This approach was adapted from the one used for the sleep deprivation of rats (*Rechtschaffen et al., 1983*). Two inactivity-dependent rotational triggers were used in this study: 220 s, which prevented the focal fly from attaining the standard definition of sleep (5 min of inactivity or more) and 1320 s, which deprived focal flies of inactivity durations of 22 min or more. Due to logistical issues relating to video-based feedback, the inactivity duration was set to 22 min – this was to make sure that no bouts of inactivity that are longer than 25 min are allowed to occur.

## Development of Ethoscope code to support yoked controls

Yoking was implemented at the level of the Ethoscope's 'TrackingUnit', which is where the connection between the video-based motion tracker and the stimulator (i.e., tube rotator) is located [self._stimulator.bind_tracker(self._tracker)]. The Ethoscope's program was modified so that the stimulator controlling the tubes containing yoked controls was bound to the tracker of their paired focal flies. This results in the stimulator of the yoked animals responding to the behavior of the focal animals at the same time as the stimulators of the focal flies. The code supporting yoked controls is available at https://github.com/shaliulab/shaferlab-ethoscope (*Ortega, 2022*; a fork of https://github.com/gilestrolab/ethoscope; *Geissmann and Gilestro, 2022*) with tag v1.0.999, which can be acquired via pip with the command pip install shaferlab-ethoscope.

## MALDI-TOF analysis

After 24 hr of sleep deprivation using 220 s immobility triggers for focal flies with their yoked and unperturbed controls, heads from all three conditions were embedded in Optimal Cutting Temperature (Tissue-Tek, Product code: 4583, Sakura Finetek USA Inc) compound, frozen with liquid nitrogen, cryo-sectioned, and prepared and assayed by MALDI-TOF MS as previously described (*Veerasammy et al., 2020*). 2,5-Dihydrobenzoc acid (Sigma-Aldrich, Cat #149357) was used as our matrix, which supports the ionization of metabolites and peptides, and is considered a useful matrix for 'universal analysis' of a diverse variety of molecular types (*Snovida et al., 2006*). MALDI mass spectra were acquired in a Bruker Autoflex Speed TOF Machine (Bruker, Germany) and peaks were identified in SCiLS MALDI Imaging software (Bruker Daltonics, Germany) by a manual peak walk across the spectrum of m/z values between 0 and 1300 for each discernible m/z ratio for unperturbed, focal, and yoked flies.

## Statistical analysis

In order to test the effect of trigger frequency and mechanical disturbance on the amount of rebound sleep, we used a two-way fixed factor ANOVA. We used trigger frequency as one fixed factor with

three levels, that is, 20 s, 120 s, and 220 s. The second fixed factor was treatment with two levels, that is, mechanically disturbed fly and unperturbed controls. Whether the difference in total sleep post-deprivation between unperturbed controls and sleep-deprived flies was dependent on trigger frequency was inferred based on the interaction effect between the two factors described above. In all other analyses reported throughout the manuscript, we either used the Mann-Whitney U (also referred to as the Wilcoxon's rank sum test) or the Kruskal-Wallis test, depending on the number of groups being compared. In all cases involving more than two groups, multiple comparisons were done using a Bonferroni correction. In case of the MALDI-TOF MS experiment, owing to the large number of peaks detected, all the p-values from individual tests were treated to a Benjamini-Hochberg correction to adjust for inflated false discovery rates. All statistical analyses were done and figures made using R (*R Development Core Team, 2022*). Specific tests, sample sizes, and p-values are reported in the figure legends.

## Acknowledgements

This work was supported by grants from the National Institute of Neurological Disorders and Stroke (R01NS077933 and R21NS131939) and start-up funds provided by the State of New York. We are grateful to Drs. Giorgio Gilestro, Bill Joiner, and Maria de la Paz Fernández for useful discussions of the work presented in this study. Dr. Ye He and Dr. Rinat Abzalimov, the Co-Directors of MALDI-TOF MS Imaging Facility, and Kelly Veerasammy at the Advanced Science Research Center, City University of New York provided technical support with MALDI-TOF experiments and imaging. Finally, we thank Matthew Ciolkowski and Robert Veline for providing useful feedback on the manuscript.

## Additional information

### Funding

| Funder | Grant reference number | Author |
| --- | --- | --- |
| National Institute of Neurological Disorders and Stroke | R21NS131939 | Orie Shafer |
| National Institute of Neurological Disorders and Stroke | R01NS077933 | Orie Shafer |

The funders had no role in study design, data collection and interpretation, or the decision to submit the work for publication.

### Author contributions

Budhaditya Chowdhury, Conceptualization, Validation, Investigation, Methodology, Writing – original draft, Writing – review and editing; Lakshman Abhilash, Conceptualization, Data curation, Software, Formal analysis, Validation, Investigation, Visualization, Methodology, Writing – original draft, Writing – review and editing; Antonio Ortega, Conceptualization, Software, Methodology; Sha Liu, Resources, Software, Supervision, Funding acquisition; Orie Shafer, Conceptualization, Resources, Supervision, Investigation, Writing – original draft, Project administration, Writing – review and editing

### Author ORCIDs

Budhaditya Chowdhury http://orcid.org/0000-0003-3836-0426
Lakshman Abhilash http://orcid.org/0000-0002-9933-8989
Orie Shafer https://orcid.org/0000-0001-7177-743X

### Decision letter and Author response

Decision letter https://doi.org/10.7554/eLife.91355.sa1
Author response https://doi.org/10.7554/eLife.91355.sa2

## Additional files

### Supplementary files
• MDAR checklist

• Supplementary file 1. Behavioral and sleep architecture phenotypes associated with mechanical sleep deprivation, and MALDI-TOF MS standards.

### Data availability
All Raw Data can be found on Dryad (https://doi.org/10.5061/dryad.qnk98sfpp).

The following dataset was generated:

| Author(s) | Year | Dataset title | Dataset URL | Database and Identifier |
|---|---|---|---|---|
| Chowdhury B, Abhilash L, Ortegan A, Liu S, Shafer O | 2023 | Data from: Homeostatic control of deep sleep and molecular correlates of sleep pressure in *Drosophila* | https://doi.org/10.5061/dryad.qnk98sfpp | Dryad Digital Repository, 10.5061/dryad.qnk98sfpp |

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
