## [Editor Report]

This important study provides a significant methodological advance for the study of *Drosophila* sleep, especially with regard to the study of its homeostatic features, as well as in its reevaluation of the 5-min rest period that is currently used to define the sleep state in *Drosophila*. Although time will tell whether the findings reported survive the test of time, this creative and imaginative piece of work provides solid strengths of evidence and food for thought, as well as important technical developments for the field.

---

## [Decision Letter]

**Decision letter after peer review:**

[Editors’ note: the authors submitted for reconsideration following the decision after peer review. What follows is the decision letter after the first round of review.]

Thank you for submitting the paper "Homeostatic control of deep sleep in *Drosophila*: Implications for Discovering Correlates of Sleep Pressure" for consideration by *eLife*.

We apologize for the delay in submitting our assessment of your manuscript but we hit some internal setbacks. Your article has now been reviewed by 3 peer reviewers, and the evaluation has been overseen by a Reviewing Editor and a Senior Editor. The following individual involved in the review of your submission has agreed to reveal their identity: Leslie C Griffith (Reviewer #4).

The reviewers were very interested in some aspects of the work, such as the use of the "yoked" protocol for assessing sleep homeostasis and the reevaluation of the 5-min rest as the defining criterion for the sleep state that is now routinely used for *Drosophila*. Nevertheless, they had a number of concerns about the data, methodology, and interpretation. Given these concerns, we are sorry to say that we cannot consider your manuscript for publication by *eLife* at this point. If you feel you can address all the issues raised, you are welcome to submit an extensively revised manuscript, which would be considered a new submission.

Recommendations for the authors

1. For this work to reach its full potential one would have to use a much larger sample base and most importantly make sure to use a wild-type line that behaves as expected when sleep-deprived. This can only be achieved by performing more experiments.

2. The most straightforward explanation for Figure 1 is that the animals are getting more short sleep bouts with the 220 s trigger.

3. It is quite notable that the baseline sleep in Figure 1 (night is almost maxed out) and later Figures (night is NOT maxed out) is quite different. This means that in those later experiments, the nighttime rebound can be seen. I think that for many data sets in the literature, this would not be the case- flies sleep a lot at night and some of your experiments are quite atypical.

4. The argument that yoked controls are better in Figure 2 is really not so clear. You still see no sleep rebound. Did you do this with the 20 or 120 triggers? It would be good to see those data especially since the 220 s trigger is the most likely to be contaminated with micro sleep and the choice to use this trigger time does not seem logical.

5. Why 24h of deprivation? Seems like this is way too stressful. Most people do not do this and I wonder if some of the failures have to do with this choice.

6. In Figure 2 almost all the recovery is at night. This will not be seen with fly lines/conditions that max out nighttime sleep. Worth at least a comment.

7. In other figures, the recovery sleep jumps around. Sometimes it is at night sometimes during the day. This really seems odd to me. I would expect it to be in one or both but consistent?

8. Should show the undisturbed condition for the 25-min sleep definition. It is like the yoked? Diff from both? This is pretty important. Overall would have liked to see more depth in this aspect of the story. I think long-bout sleep could be really interesting and useful as an alternative paradigm.

9. Did you look at PWake/PDoze for your data sets? This analysis might help shore up some of the arguments you make about sleep pressure.

10. To increase the value of this study, I might encourage the authors to extend their MALDI-TOF work to its logical conclusion: what are the molecules that have significantly increased because of sleep pressure? Can they be identified (and replicated)? Do they make sense with regard to other studies of deep sleep function? Also, I might encourage the authors to expand their sleep analyses to short-sleep epochs as well, meaning 1-5min. If >25min inactivity bouts appear to engage 'deep sleep' functions, it is also possible that very short sleep bouts (1-5min), which have always been excluded from analysis in most *Drosophila* sleep studies, might engage 'active sleep' functions, which could also be under homeostatic control. While it is commendable that the authors question the 5min sleep criterion (Figure 4B is especially informative in that regard, and a version of this figure might be provided for ALL the deprivation protocols), they still stick to this threshold on the short end of their data. Any immobility over 1min is probably worth investigating, as has been shown by a number of video-based studies now.

11. A final small correction: in line 108 the authors state that an SWS-like signal was found in flies, ref. 50 and 51. This is incorrect. No SWS-like signal has ever been found during sleep in fly brain activity, so far. In the studies referenced (50 and 51), only a decrease in LFP amplitude was observed. Some recent studies (e.g., Raccuglia et al. 2019 and 2022) have identified ~1Hz oscillatory neural activity in the central brain of sleep-deprived flies, but it is unclear if these flies were sleeping or simply sleep-deprived and awake. To imply that slow wave activity is a prerequisite for 'deep sleep' in any animal is misguided. Deep sleep (or active sleep) should be defined by its functions, not by the peculiar electrophysiology of certain animals, such as the minority of animals endowed with a cortex (or eyeballs).

[Editors’ note: further revisions were suggested prior to acceptance, as described below.]

Thank you for submitting the revised version of the paper "Homeostatic control of deep sleep in *Drosophila*: Implications for Discovering Correlates of Sleep Pressure" for consideration by *eLife*. Your article has been re-reviewed by 2 of the original 3 peer reviewers (reviewer #4 was unavailable at this time), and the evaluation has been overseen by a Reviewing Editor and a Senior Editor (Claude Desplan).

Both reviewers pointed out that the manuscript has been much improved. However, there are a couple of remaining issues that need to be addressed, as outlined below:

Comments to authors:

This Important study provides a significant methodological advance for the study of *Drosophila* sleep, especially with regard to the study of its homeostatic features as well as in its reevaluation of the 5-min rest period that is currently used to define the sleep state in *Drosophila*. The revised version of the manuscript addressed most of the shortcomings originally identified by the reviewers. Nevertheless, they have identified a couple of issues that remain problematic, and which would need to be suitably addressed in order for this solid work to be considered a benchmark study.

In this study, Chowdhury and co-authors use Ethoscope to re-evaluate key approaches and metrics that have dominated the *Drosophila* sleep field over the past two decades. The first approach involves mechanical sleep deprivation techniques, where sleep-deprived flies are typically compared to controls that have not been mechanically stimulated. The authors rightly question the appropriateness of such controls, and adapt Ethoscope to provide instead yoked controls that are equally stimulated but can achieve sleep. They find that the yoked control approach yields marginally better sleep rebound data over successive days following the sleep deprivation procedure. They then find that sleep rebound data is significantly improved if only >25min sleep bouts are examined. This suggests that their sleep deprivation protocol specifically impacts longer sleep bouts, rather than shorter sleep bouts defined by the traditional 5-min criterion. They posit that this is because sleep homeostasis in flies (like in mammals) is centered on deep sleep functions, which are more fully engaged during longer sleep bouts. Finally, to support their case in a different way, the authors employ MALDI-TOF (a technique which allows quantifying protein content differences in tissue sections) to show that most of the protein changes incurred during a mechanical sleep deprivation procedure result from mechanical stress rather than sleep pressure. They employ the same three comparisons (unstimulated, yoked controls, sleep deprived) to arrive at this important conclusion. They uncover only 5 proteins in the fly central brain with increased expression due to sleep pressure, a small minority compared to the number of proteins associated with stress alone. The authors make a strong case for the field to re-evaluate the way sleep is being studied in this model, suggesting that many studies examining the homeostatic regulation of sleep in *Drosophila* might have been confounded by the effects of stress, or by the examining too short sleep epochs.

The main strength of this study is their clever usage of Ethoscope to better understand sleep homeostasis in the fly model. The application of yoked controls has been a standard in the mammalian sleep field and requires video-based methods linked to individualized feedback in order to apply similar techniques to flies. Everything follows from there: fine-tuning the number of random stimulations, exploring different inactivity durations to call a fly asleep, comparing yoked controls versus unstimulated controls in a variety of scenarios, including MALDI-TOF. Thus, the manuscript describes a welcome attempt at redefining sleep rebound in *Drosophila melanogaster* using some original or historically re-discovered definitions which are, in principle, useful for the field as a whole.

We thank the authors for the extensive revision of the manuscript, which addressed most of the shortcomings originally identified by the reviewers. This is an important study, which will benefit the *Drosophila* sleep community and inform the broader sleep field. Nevertheless, there remain a couple of issues that still need to be suitably addressed. That being said, this is a nice paper!

1. The main issue that is unclear is the one presented in L325-328 and Figure 2H-K. It is really not intuitive how this result is possible: no rebound is observed when comparing sleep deprived flies to rested, undisturbed flies. Yet a rebound is observed when comparing SD flies to flies that experienced partial sleep disruption. How could "sleep-independent effects of mechanical perturbation mask homeostatic sleep responses" as proposed in L327? This conundrum is not appropriately addressed in the discussion.

2. Figure 2A-D – Please note that using a similar paradigm Geissmann et al. did see SD rebound after 220s SD for 12h (Figure 5). If correct, this discrepancy with the existing literature should be discussed and referred to at ~L605 especially considering the similarity in the methodology used in both studies.

3. Figures 2 and 3: The red line in Figure 2H is identical to the red lines in Figures 3A and 3B. The blue line in Figure 2H is the same as in Figure 3B. Although this is a rather large dataset (n=82) if the use of the same data is strictly necessary, then it is important to point out in the text and figures that this is the same experimental dataset used in different visualizations and comparisons because the readers may otherwise get an incorrect illusion of reproducibility. We would recommend either putting all these results in the same panel/figure or independently repeating the experiment shown in Figure 3.

4. Figure 4A. The risk with this kind of analysis is that it does not take into account pre-existing variability. If a 3 day period is to be analyzed after SD, statistically speaking it would be better to include a longer baseline (pre-SD) too. For instance, the focal flies do show a sleep peak ZT12 of the day before SD. Also, for analysis of what is proposed to be sleep depth, we would recommend giving a chance to the Hidden Markov Chain model from the Griffith lab beyond what is done and shown in Supplementary Figure 3.

5. Figure 5: please mark on the figure what is the day of SD/treatment.

6. Line 662: "We replicated…". Does this mean that the MALDI-TOF experiment was an N=1 (one fly brain) for each condition, so 'replicated' means that a second fly was examined per condition? This seems quite thin and might need to be explicitly stated (also in the methods), with some caveats about this being quite a preliminary finding.

7. Line 576 mentions an unknown molecule. It might nevertheless be interesting to canvass some candidates, especially since this seems to have been replicated.

8. Line 617 (and elsewhere): It should be emphasized that 1-5min sleep could in principle also be 'deep', especially after sleep deprivation. This is indeed what van Alphen 2013 showed, that after SD flies descended more rapidly into deeper (less arousable) sleep, even within the first 5min. So rebound deep sleep could still be happening within the 220 s deprivation protocol. It might be helpful to discriminate between 'short sleep' which is any 1-5min sleep epoch, and 'active sleep' which can really only be confirmed with additional investigation (e.g., responsiveness, brain recordings). We would recommend that the authors use the term 'short sleep' when discussing <5min sleep bouts, and 'active sleep' only as an interpretation of one of the kinds of sleep likely to occur during short sleep bouts, especially in non sleep-deprived flies.

9. The point of Supplementary Figure 2 is a little hard to follow. Sleep-deprived flies are more active at night compared to controls, but not when yoked. This might need some clarifying, as it's somewhat unexpected and refers to a separate issue from the one being discussed at this point in the paper.

---

## [Author Response]

[Editors’ note: the authors resubmitted a revised version of the paper for consideration. What follows is the authors’ response to the first round of review.]

Recommendations for the authors1. For this work to reach its full potential one would have to use a much larger sample base and most importantly make sure to use a wild-type line that behaves as expected when sleep-deprived. This can only be achieved by performing more experiments.

To address this concern, we have expanded our analysis of sleep rebound following mechanical deprivation using the widely used vortexer method and have replicated sleep homeostasis experiments using the standard 20s frequency across a range of deprivation durations and show that our wild-type flies behave as described in previous work. In addition, we have extended our deprivation experiments to include one of the most commonly used genetic backgrounds to confirm that we see patterns of sleep rebound in this strain that are similar to the ones reported for our wild-type strain. Thus, we have shown that our surprising results with 220s inactivity triggers, which produce the same level of sleep deprivation as the 20s triggers, are not due to an idiosyncrasy with our wild-type strain. This expanded analysis strengthens our conclusion that, when using standard methods, behavior following sleep deprivation tracks mechanical perturbation rather than sleep pressure.

2. The most straightforward explanation for Figure 1 is that the animals are getting more short sleep bouts with the 220 s trigger.

To address this alternative explanation, we have examined short, “active” sleep bouts (inactivity lasting between one and five minutes) following our sleep homeostasis experiments. It is indeed true that flies sneak in short sleep bouts during the 220s deprivation window. This, therefore, does raises the possibility that flies are not accruing sufficient sleep pressure to show rebound post sleep deprivation, which we now acknowledge in the text. Remarkably, we find such increases in active sleep even when using 20-s trigger frequencies for mechanical deprivation, a condition that does result in rebound:

**Author response image 1. sa2fig1:** 

Our subsequent experiments in which we deprive flies of only long bout sleep (22-min trigger frequency) and find significant multi-cycle long bout rebound is strong evidence that long-bout sleep is potently controlled by homoestatic mechanisms (also discussed in the updated text). This has not been previously shown.

3. It is quite notable that the baseline sleep in Figure 1 (night is almost maxed out) and later Figures (night is NOT maxed out) is quite different. This means that in those later experiments, the nighttime rebound can be seen. I think that for many data sets in the literature, this would not be the case- flies sleep a lot at night and some of your experiments are quite atypical.

We agree that we could have been clearer for the reasons behind this striking discrepancy across experiments, which are due to the use of different methods to record behavior (video versus beam crossing) and a turn to the examination of long-bout sleep (inactivity durations of 25 minutes or more rather than 5 minutes or more). In our revised manuscript we describe this more clearly and provide the reasons for these discrepancies. It is important to note that, when using video analysis or multi-beam DAM recordings, flies do not max out on the amount of night-time sleep. We are therefore confident that our results are not atypical to laboratory conditions or fly strains but reflect the true nature of night-time sleep.

4. The argument that yoked controls are better in Figure 2 is really not so clear. You still see no sleep rebound. Did you do this with the 20 or 120 triggers? It would be good to see those data especially since the 220 s trigger is the most likely to be contaminated with micro sleep and the choice to use this trigger time does not seem logical.

We have provided a clearer rational for using the 220s trigger in our revised manuscript. As described in our response to concern number two above, we do find increased microsleep during the deprivation window as pointed out by the reviewer. However, our subsequent experiments and analyses show that the homeostatic aspects of sleep rebound are captured most sensitively by long bouts of sleep, and establish robust homeostatic rebound in long, deep sleep rebounds, despite the presence of brief instances of inactivity during deprivation using 220-s inactivity triggers. Furthermore, 220s inactivity triggers were chosen to prevent the attainment of the sleep state as currently defined (widely accepted as inactivity of 5-min or more for over 20 years) in our focal flies while allowing for substantial sleep in our yoked controls. Shorter inactivity triggers would not have allowed our yoked flies to sleep much at all, thereby obviating their u9lity. We have made this fact clearer in the new manuscript.

5. Why 24h of deprivation? Seems like this is way too stressful. Most people do not do this and I wonder if some of the failures have to do with this choice.

The 24h duration of sleep deprivation was chosen to avoid circadian confounds and to examine sleep homeostasis under conditions more akin to those used in mammalian studies. It is true that most inves9gators of *Drosophila* sleep have se;led on shorter (12h, 6h, or 4h) durations of deprivation. However, 24h deprivation has been used in the fly previously (e.g., Huber et al. 2004 Sleep Vol. 27, p. 628; Geissman et al. 2019 Sci. Adv. Vol. 5, eaau9253). Applying mechanical stimuli at specific times within the circadian cycle is expected to reset the circadian clock (e.g., Simoni et al. 2014 Science Vol. 343, p. 525). Given the clock’s strong influence on sleep, this represents an additional confound of mechanical sleep deprivation. We therefore opted to deliver mechanical stimuli for deprivation across the en9re diurnal cycle to avoid the circadian phase responses that would be expected from stimuli delivered at specific times within the cycle. Though 24h of deprivation is not commonly used in the fly field, the loss of one day’s sleep is not an extraordinary amount of sleep deprivation in the context of the wider field of sleep science. Human subjects have been routinely kept awake for significantly longer than 24h and a large number of studies have deprived mammalian model systems of full cycles of sleep. Furthermore, previous work using the ethoscope system has established that flies can be deprived of sleep for up to at least 40 days without a significant increase in mortality, which would not be expected if flies were being subjected to overwhelming stress.

However, as described above, we have expanded our analysis to include shorter durations of deprivation. In doing so, we find that, when using standard approaches to sleep deprivation and the quantification of rebound, 24h of sleep deprivation on the vortexer fails to produce substantially larger rebounds than 12h or 6h deprivation. This further supports our contention that standard methods in the field do not offer a sensitive means of detecting *bona fide* sleep rebound, as longer durations of deprivation produce greater amounts of sleep loss and should therefore produce larger homeostatic rebounds. This is clearly not the case using the most widespread approach to sleep homeostasis in the fly and calls for a reexamination of the methods employed by the field.

6. In Figure 2 almost all the recovery is at night. This will not be seen with fly lines/conditions that max out nighttime sleep. Worth at least a comment.

As discussed in our response to concern three above, sleep measurements using the ethoscope do not display maxed-out sleep the way that single-beam crossing data do. This is due to the overestimation of inactivity and sleep produced by the single-beam monitoring provided by the Trikinetics DAM system (see Geissman et al. 2019 Sci. Adv. Vol. 5, eaau9253), rather than due to idiosyncrasies of our flies or environmental conditions. We have made this point clearer in our revised manuscript.

7. In other figures, the recovery sleep jumps around. Sometimes it is at night sometimes during the day. This really seems odd to me. I would expect it to be in one or both but consistent?

We are struck by this too and note that the picture becomes much more consistent when analysis is focused on long-bout sleep, which displays homeostatic increases in both daytime and – sleep (compared to yoked controls) across several diurnal cycles. Thus, by examining longer, deeper sleep and controlling for the sleep-independent effects of mechanical deprivation, we observe a much simpler response to deprivation, in which sleep is increased at all times of the day during which normal sleep occurs and across several days following deprivation. We think that this provides further evidence for the u9lity and power of our new approach. We have revised our manuscript to make this point clear to the reader.

8. Should show the undisturbed condition for the 25-min sleep definition. It is like the yoked? Diff from both? This is pretty important. Overall would have liked to see more depth in this aspect of the story. I think long-bout sleep could be really interesting and useful as an alternative paradigm.

Though we did report sleep in the unperturbed controls in our original manuscript, we agree that the temporal pattern of long-bout sleep is an important point that deserves more attention. We now provide a more detailed description of these points in the revised manuscript and show that long bout sleep time-series are remarkably uniform in control flies across multiple experimental runs.

9. Did you look at PWake/PDoze for your data sets? This analysis might help shore up some of the arguments you make about sleep pressure.

We have added this analysis to our study and find that focally sleep-deprived flies display significant increases in PDoze compared to yoked controls across three diurnal cycles following deprivation, further supporting our contention that the use of yoked controls and the examination of long duration sleep bouts reveals a large and persistent sleep pressure that must be gradually dissipated following deprivation.

10. To increase the value of this study, I might encourage the authors to extend their MALDI-TOF work to its logical conclusion: what are the molecules that have significantly increased because of sleep pressure? Can they be identified (and replicated)? Do they make sense with regard to other studies of deep sleep function? Also, I might encourage the authors to expand their sleep analyses to short-sleep epochs as well, meaning 1-5min. If >25min inactivity bouts appear to engage 'deep sleep' functions, it is also possible that very short sleep bouts (1-5min), which have always been excluded from analysis in most *Drosophila* sleep studies, might engage 'active sleep' functions, which could also be under homeostatic control. While it is commendable that the authors question the 5min sleep criterion (Figure 4B is especially informative in that regard, and a version of this figure might be provided for ALL the deprivation protocols), they still stick to this threshold on the short end of their data. Any immobility over 1min is probably worth investigating, as has been shown by a number of video-based studies now.

We have replicated our unbiased screen for molecular correlates of sleep pressure and show that serotonin, a biogenic amine that has long been implicated as a sleep promoting substance in both mammalian and fly brains, meets the criteria for a sleep substance within the central brain of *Drosophila*. This finding would appear to reconcile conflicting observations regarding serotonin’s role in sleep regulation (see Jouvet 1999 *Neuropsychopharmacology* Vol. 21, p. 24). We think that this result will be of major interest to the field and will be the basis of a great deal of future work. While we find the reviewer’s comments regarding our re-evaluation of sleep definition in the fly encouraging, we must reiterate that although we agree ‘active sleep’ likely represents a discrete sleep stage in the fly that is likely relevant to sleep homeostasis, it would not be possible to examine it with yoked controls simply owing to the logistical issues associated with yoking (i.e., the frequency of simulation necessary to prevent it; see figure above) would make it impossible for yoked controls to get enough sleep to serve as meaningful controls for mechanical stimulation. Nevertheless, we agree that future studies should address the important question of homeostatic control of the active sleep state.

11. A final small correction: in line 108 the authors state that an SWS-like signal was found in flies, ref. 50 and 51. This is incorrect. No SWS-like signal has ever been found during sleep in fly brain activity, so far. In the studies referenced (50 and 51), only a decrease in LFP amplitude was observed. Some recent studies (e.g., Raccuglia et al. 2019 and 2022) have identified ~1Hz oscillatory neural activity in the central brain of sleep-deprived flies, but it is unclear if these flies were sleeping or simply sleep-deprived and awake. To imply that slow wave activity is a prerequisite for 'deep sleep' in any animal is misguided. Deep sleep (or active sleep) should be defined by its functions, not by the peculiar electrophysiology of certain animals, such as the minority of animals endowed with a cortex (or eyeballs).

We regret this error in our scholarship and have corrected this in our revised manuscript.

[Editors’ note: what follows is the authors’ response to the second round of review.]

Both reviewers pointed out that the manuscript has been much improved. However, there are a couple of remaining issues that need to be addressed, as outlined below:Comments to authors:This Important study provides a significant methodological advance for the study of Drosophila sleep, especially with regard to the study of its homeostatic features as well as in its reevaluation of the 5-min rest period that is currently used to define the sleep state in *Drosophila*. The revised version of the manuscript addressed most of the shortcomings originally identified by the reviewers. Nevertheless, they have identified a couple of issues that remain problematic, and which would need to be suitably addressed in order for this solid work to be considered a benchmark study.In this study, Chowdhury and co-authors use Ethoscope to re-evaluate key approaches and metrics that have dominated the *Drosophila* sleep field over the past two decades. The first approach involves mechanical sleep deprivation techniques, where sleep-deprived flies are typically compared to controls that have not been mechanically stimulated. The authors rightly question the appropriateness of such controls, and adapt Ethoscope to provide instead yoked controls that are equally stimulated but can achieve sleep. They find that the yoked control approach yields marginally better sleep rebound data over successive days following the sleep deprivation procedure. They then find that sleep rebound data is significantly improved if only >25min sleep bouts are examined. This suggests that their sleep deprivation protocol specifically impacts longer sleep bouts, rather than shorter sleep bouts defined by the traditional 5-min criterion. They posit that this is because sleep homeostasis in flies (like in mammals) is centered on deep sleep functions, which are more fully engaged during longer sleep bouts. Finally, to support their case in a different way, the authors employ MALDI-TOF (a technique which allows quantifying protein content differences in tissue sections) to show that most of the protein changes incurred during a mechanical sleep deprivation procedure result from mechanical stress rather than sleep pressure. They employ the same three comparisons (unstimulated, yoked controls, sleep deprived) to arrive at this important conclusion. They uncover only 5 proteins in the fly central brain with increased expression due to sleep pressure, a small minority compared to the number of proteins associated with stress alone. The authors make a strong case for the field to re-evaluate the way sleep is being studied in this model, suggesting that many studies examining the homeostatic regulation of sleep in Drosophila might have been confounded by the effects of stress, or by the examining too short sleep epochs.The main strength of this study is their clever usage of Ethoscope to better understand sleep homeostasis in the fly model. The application of yoked controls has been a standard in the mammalian sleep field and requires video-based methods linked to individualized feedback in order to apply similar techniques to flies. Everything follows from there: fine-tuning the number of random stimulations, exploring different inactivity durations to call a fly asleep, comparing yoked controls versus unstimulated controls in a variety of scenarios, including MALDI-TOF. Thus, the manuscript describes a welcome attempt at redefining sleep rebound in *Drosophila melanogaster* using some original or historically re-discovered definitions which are, in principle, useful for the field as a whole.We thank the authors for the extensive revision of the manuscript, which addressed most of the shortcomings originally identified by the reviewers. This is an important study, which will benefit the *Drosophila* sleep community and inform the broader sleep field. Nevertheless, there remain a couple of issues that still need to be suitably addressed. That being said, this is a nice paper!1. The main issue that is unclear is the one presented in L325-328 and Figure 2H-K. It is really not intuitive how this result is possible: no rebound is observed when comparing sleep deprived flies to rested, undisturbed flies. Yet a rebound is observed when comparing SD flies to flies that experienced partial sleep disruption. How could "sleep-independent effects of mechanical perturbation mask homeostatic sleep responses" as proposed in L327? This conundrum is not appropriately addressed in the discussion.

The reviewers here have identified the paradoxical nature of fly sleep deprivation studies that we think deserves serious attention within the field. It is indeed surprising (and troubling), that sleep deprived flies often do not display easily detectable sleep rebound when compared to undisturbed flies. This is particularly clear when flies are deprived of sleep for a full 24-h cycle (Figure 1). We believe that this absence of rebound is related to the relatively small rebounds that are measured using the standard approaches in the field. We agree that this produces a conundrum for the field that must be addressed. We contend that identifying this conundrum is one of the important contributions of our study. Even without a full answer to this conundrum our results show that the standard practice of attributing any behavioral/molecular differences observed between frequently perturbed flies to unperturbed flies to differences in sleep pressure alone is problematic and seriously confounded. Sleep-independent effects of any method of deprivation are likely multifactorial. The utility of the yoked control paradigm introduced by Rechtschaffen is that these effects do not need to be completely understood in order to be accounted for. Though it is not within the scope of our study to completely account for the sleep-independent effects of mechanical deprivation, we suggest a partial explanation: the depriving stimuli itself causes increases in locomotor activity in deprived flies that likely antagonize the increased drive to sleep in deprived flies (Supplementary Figure 2), which is related to comment 9 below. We have expanded this point further in lines 331-335, 338-339, 340-341, and 609-615.

2. Figure 2A-D – Please note that using a similar paradigm Geissmann et al. did see SD rebound after 220s SD for 12h (Figure 5). If correct, this discrepancy with the existing literature should be discussed and referred to at ~L605 especially considering the similarity in the methodology used in both studies.

The difference in observed rebound in our approach and that of Geissmann et al., likely arises from two sources. The first is that Geissman deprived flies for 12-h whereas we did for 24-h. Our work using vortexers and Trikinetics monitors (Figure 1) establish the paradoxical observation that longer durations sleep deprivation can produce smaller rebounds (or no rebound) compared to briefer durations. Second, we employed a different rebound analysis window (3 hours in Geissman et al. and 24 hours in our study) and a distinct metric for quantification of rebound. We have described this discrepancy in lines 667-676.

3. Figures 2 and 3: The red line in Figure 2H is identical to the red lines in Figures 3A and 3B. The blue line in Figure 2H is the same as in Figure 3B. Although this is a rather large dataset (n=82) if the use of the same data is strictly necessary, then it is important to point out in the text and figures that this is the same experimental dataset used in different visualizations and comparisons because the readers may otherwise get an incorrect illusion of reproducibility. We would recommend either putting all these results in the same panel/figure or independently repeating the experiment shown in Figure 3.

We separated the plots for the sake of visual clarity in these cases. Since we are reporting data from the same experiments, we think it necessary to re-use the plots when profiles are separated in this way. We now make it clear in the figure legend that Figure 3B is an extended analysis of the same data reported in Figure 2H.

4. Figure 4A. The risk with this kind of analysis is that it does not take into account pre-existing variability. If a 3 day period is to be analyzed after SD, statistically speaking it would be better to include a longer baseline (pre-SD) too. For instance, the focal flies do show a sleep peak ZT12 of the day before SD. Also, for analysis of what is proposed to be sleep depth, we would recommend giving a chance to the Hidden Markov Chain model from the Griffith lab beyond what is done and shown in Supplementary Figure 3.

We agree that accounting for variability in baseline sleep, may have revealed small amounts of rebound that our approach did not. However, we do not think that this would have fundamentally changed the major results of our study. For example, the relatively large rebounds we measure when accounting for the effects of mechanical deprivation (through yoked controls) and focusing analysis on longer bouts of sleep (by changing inactivity criterion for sleep) would still be striking contrast with the small amounts of rebound typically observed only within short windows following deprivation. Likewise, the inclusion of more baseline days may have changed the magnitudes of our measured rebounds, but having the deprivation occur several days into behavioral recording would introduce an additional difference between our study and most previous studies, i.e., the age of flies during sleep deprivation, which would have made it challenging for us to compare our results to previous studies. We would also like to stress that although the focal flies seem to have higher sleep at a time point just after ZT12 before sleep deprivation, the total amount of sleep over a period of 24-h is not statistically significantly different between treatments.

We claim that the sleep pressure of individual flies, persists for at least three cycles post sleep deprivation. According to Wiggin et al., 2020 p(Doze) is considered a measure of sleep pressure, which is precisely why that was quantified to make our case. Importantly, the authors of the paper write, “These hidden [Markov] states have a predictable relationship with P(Doze) and P(Wake), suggesting that the methods capture the same behaviors.” The authors go on to also write, “Overall, the HMM approach recapitulates the results of the P(Wake)- and P(Doze)-based analysis with the benefit of allowing decoding of behavior into discrete categories that may be tied to different circuit configurations or balances between circuits.” Moreover, there is no established association between the different hidden sleep stages and length of sleep bouts. Therefore, we think that although useful and strong as a method, the addition of HMM analyses to our dataset would not change the major results/conclusions of our study and would take us significant time.

5. Figure 5: please mark on the figure what is the day of SD/treatment.

We have now marked on Figure 2 – 5 the exact 24-hour duration of SD/treatment.

6. Line 662: "We replicated…". Does this mean that the MALDI-TOF experiment was an N=1 (one fly brain) for each condition, so 'replicated' means that a second fly was examined per condition? This seems quite thin and might need to be explicitly stated (also in the methods), with some caveats about this being quite a preliminary finding.

The replicated MALDI-TOF was not done with only one head per sample. We now report the exact number of heads used for each condition in the figure legend (line 1128-1130).

7. Line 576 mentions an unknown molecule. It might nevertheless be interesting to canvass some candidates, especially since this seems to have been replicated.

There is a wide range of possibilities for a molecule of this size. Narrowing down these possibilities will require further extensive experimentation that is beyond the scope of this study, and we prefer not to speculate. However, we discuss the kinds of molecules that would be represented by this m/z value in the revised manuscript (lines 583-587).

8. Line 617 (and elsewhere): It should be emphasized that 1-5min sleep could in principle also be 'deep', especially after sleep deprivation. This is indeed what van Alphen 2013 showed, that after SD flies descended more rapidly into deeper (less arousable) sleep, even within the first 5min. So rebound deep sleep could still be happening within the 220 s deprivation protocol. It might be helpful to discriminate between 'short sleep' which is any 1-5min sleep epoch, and 'active sleep' which can really only be confirmed with additional investigation (e.g., responsiveness, brain recordings). We would recommend that the authors use the term 'short sleep' when discussing <5min sleep bouts, and 'active sleep' only as an interpretation of one of the kinds of sleep likely to occur during short sleep bouts, especially in non sleep-deprived flies.

We have changed “active sleep” to “short-sleep” bouts when describing data in the revised manuscript. We now refer to “active sleep” only when discussing the work of Anthony et al., 2023.

9. The point of Supplementary Figure 2 is a little hard to follow. Sleep-deprived flies are more active at night compared to controls, but not when yoked. This might need some clarifying, as it's somewhat unexpected and refers to a separate issue from the one being discussed at this point in the paper.

These data were included in our revised manuscript to address the issues raised in point one above. Mechanical perturbation causes an increase in locomotor activity in mechanically deprived flies compared to unperturbed controls. This would be expected to mask, at least partially, the increased sleep drive expected of deprived flies. Please see our response to comment 1 above.